# Transformations, trajectories, and similarities of national production structures: A comparative fingerprinting approach

**Carl Nordlund**[1,2]*

**1** Department of Management and Engineering, Institute for Analytical Sociology, Linköping University, Linköping, Sweden, **2** Department of Economic History, Lund University, Lund, Sweden

* carl.nordlund@liu.se

## Abstract

This article proposes a network-analytical framework for the comparative study of national production structures in global production networks. Conceptualizing such structures as the linked networks of both domestic and foreign intermediate inputs, the latter constituting the characteristic feature of contemporary economic globalization, the proposed approach extracts a structural profile that captures the up- and downstream prominence of economic sectors for a particular country and year. These 'fingerprints' of national production structures can subsequently be compared on a pairwise basis, providing novel ways to determine and compare the structural similarities, transformations, and trajectories of national economies in the transnational production regime. Two shorter case studies exemplify the approach. The first applies clustering methods to explore spatiotemporal similarities of the production structures for 40 countries over the 1995–2011 period. Based on such similarities, an analytically useful classification into 11 structural types is proposed. The second study addresses structural transformations and trajectories during EU's eastern enlargement, finding significant structural change, yet minuscule East-West convergence.

## 1. Introduction

Breaking with the classical and neoclassical narratives of economic globalization, in which commodities produced within singular nations are traded on international markets, recent decades have seen a substantial increase in the trans-nationalization of production itself. Gereffi's now decades-old automobile example continues to illustrate this well: a 1994 Ford Escort was produced in 15 countries, spanning three continents, extending the archetypal assembly lines of Fordism across the globe [1, also 2, 3]. Indeed, the existence of global commodity chains and fragmented production is not a novel phenomenon, having precursors traced back to 'the long 16th century' [4], but the intensity, scope and systemic nature of this 'second unbundling of globalization' is unprecedented [5, 6]. This ongoing international integration of national systems of production has had, and will most likely continue to have, a profound impact on the developmental trajectories of nations, and thus also on the specific narratives,

**Data Availability Statement:** All data used in this article is publicly available for download at the project website: https://www.demesta.com/fingerprinting/.

**Funding:** This research was supported by NordForsk through the funding to The Network

Dynamics of Ethnic Integration (project number 105147), the Swedish Research Council (445-2013-7681), and Central European University Budapest Foundation (CEU BPF). The funders had no role in study design, data collection and analysis, decision to publish, or preparation of the manuscript. There was no additional external funding received for this study.

**Competing interests:** The author has declared that no competing interests exist.

models and analytical approaches we employ to understand such dynamics at the national level [7–11].

To map and compare how national production structures transform and evolve within this transnational production regime, several different analytical approaches are at our disposal, each representing a specific take on how such structures can be conceptualized and operationalized. Among the more frequently applied approaches, two broad schools have emerged, both sharing the common aim of capturing and comparing features of national production systems albeit from different conceptual points of view.

The first and fairly recent contribution to the analytical toolbox for understanding national production is the 'economic complexity' framework proposed by Hidalgo and colleagues [12, 13]. With a stated aim to "quantify the complexity of a country's economy" [13], where the placement and trajectory of a nation in the generated 'production space' "can be used to analyze the evolution of a country's productive structure" [12], their proposed approach and metrics for such analyses are solely based, by design [14], on compositions of national gross exports. Export commodity bundles and their similarities across nations could indeed be interesting [15–17], but the interpretation of transformed country-product bipartite data is arguably more relevant under the assumption that national gross exports have been fully produced within these countries, which is contrary to the defining feature of transnational production [17–22].

Building instead on these more contemporary ideas about fragmented production, a large body of work studies the creation and trade in value-added of sectors and nations [7, 23–27]. As the creation and trade in value-added indeed both shape and are shaped by national production structures, such analyses should indeed reflect certain aspects of the properties and transformations of the internal production structures of nations, especially when also distinguishing between different types of value-added activities [17].

Blurring the separation between these two broad approaches, combining the computational approach of economic complexity's product space with value-added export data, the 'industry space' captures similarities in sectorial value-added exports between national economies [22, 28]. The industry space approach is, similar to its product space sibling, aimed at comparing economic structures of nations over time: economies with similar sectoral distributions of their value-added exports would thus, it is argued, reflect a similarity with respect to the capabilities of national production structures.

Temporal and international comparisons of sectorial and functional distributions and trade in value-added, as well as gross or value-added export compositions, undoubtedly provide key insights on the structure and dynamics of the contemporary transnational production regime and its national components. However, if the intention of such analyses is to compare national production structures with each other, such comparisons are at best by proxy. Country-specific gross or value-added exports might indeed be *associated* with the production structures of a national economy, output whose compositions surely reflect the particular industrial structures and configurations that yield such outputs, but neither of these types of outputs *are* said production structures. Rather, for more direct analyses and comparisons of national production structures, we should arguably shift our attention to what goes on "under the hood" [29, also 30] of national economies–i.e. the properties of the complex networks of interdependent flows between and within economic sectors that *result* in the value-added, export compositions, and gross domestic output that the conventional comparative metrics and measures capture.

This paper proposes a novel analytical framework for the comparative study of national production structures within global production networks. Extending and adapting an earlier eigenvector-based approach [31] for the contemporary era of transnational production, here

conceptualizing a national production structure as the complex network of economic flows between and within domestic economic sectors that also includes foreign intermediate inputs, the proposed approach extracts a structural profile in terms of up- and downstream sectorial prominence for a given country and year. Having determined such 'fingerprints' of national production, these can subsequently be compared in a way that is theoretically and conceptually grounded, computationally transparent, and, in the context of global production networks, arguably and hopefully demonstrated to be both useful and insightful when studying structural similarities, transformations and trajectories of national economies.

Using national input-output data for 40 countries from the WIOD 2013 dataset [32] between 1995–2011, the usefulness of the proposed framework is demonstrated through two short case studies.

The first case study explores similarity patterns in the full set of extracted fingerprints. Using agglomerative hierarchical clustering, an analytically useful classification into 11 structural types is proposed. Tracking how countries transition between these types, several such transitions seem to coincide with specific economic-historical events of respective country. As a contrast to these findings, a corresponding cluster analysis based on gross sectorial exports is conducted, yielding clusters that do not capture the same structural distinctions caught by the proposed fingerprinting approach.

The second case study tracks structural transformation of European economies during EU's eastern expansion. Tying into pertinent questions on divergent developmental pathways of East European nations following their accession [33], this case study demonstrates longitudinal approaches to comparative fingerprint analysis as means to examine structural transformations and would-be convergences among five Western and ten Central- and East-European countries. Finding significant structural change among these Eastern economies, their developmental trajectories were more of an orbital kind: indeed on the move, yet seemingly remaining equidistant to the relatively more stable Western production structures.

A project website (https://demesta.com/fingerprinting) supplements this article and provides access to the complete set of WIOD-derived fingerprints. The website also hosts implementations of the various analytical tools presented in this article, allowing for additional analyses and explorations not covered here (figures marked with "[D3js]" were created using the set of interactive components available on the project website). An appendix also supplements this article, providing both auxiliary tables and data as well as some additional analyses and comparisons with alternative methods and metrics.

The next section provides a brief introduction to national input-output tables, highlighting the segments that arguably correspond to the production structure of an economy. A brief overview of previous approaches for comparing such structures follows, emphasizing their shortcomings in the context of transnational production. The fingerprint approach is then specified, followed by examples of both individual and comparative fingerprints. Two eigenvalue-based diagnostic measures are introduced, leading to the removal of 27 of the 680 country-year combinations in the WIOD13 dataset. The two case studies follow, and a short summary and outline for future research conclude this article.

## 2. Fingerprinting national input-output tables

A national input-output table is an account of economic flows that occur between different parts of a national economy, and to different ends, during a specific time. With early precursors in the work of Quesnay and Walras, it was Wassily Leontief's work that popularized the approach [34–36]. In 1951, Walter Isard extended the idea of input-output analysis by combining multiple regional (single-economy) tables together into inter-regional input-output tables

**Table 1. The structure of a national input-output table (from WIOD13).**

| | | Sector | | | Final use | | | Total |
|---|---|---|---|---|---|---|---|---|
| | | $c_1$ | .. | $c_{34}$ | (Various end uses) | | | |
| Sector | $c_1$ | Intermediate use (Z) | | | Domestic final use (DFU) | Exports (E) | | Total output (x) |
| | .. | | | | | | | |
| | $c_{34}$ | | | | | | | |
| Sector | $c_1$ | Imports (M) | | | Imports final use (IFU) | | | |
| | .. | | | | | | | |
| | $c_{34}$ | | | | | | | |
| | | Value added (VA) | | | | | | |
| | | Total output (x) | | | | | | |

[37]. More recent advances in data processing and the international harmonization of economic statistics have resulted in several Isard-style multi-regional input-output datasets, such as the World Input-Output Database (WIOD) [32].

Representing a typical national input-output table, Table 1 below depicts the structure of a national table in the WIOD dataset (2013 release). The submatrix Z contains the valued directional flows between and within 34 domestic sectors, this being the "largest and, for most empirical analyses, the most important part of the table" [38]. In addition to intermediate inputs originating from domestic sectors (i.e. the Z matrix), such upstream flows can also have foreign origins, the latter represented by submatrix M. Whereas the full inter-regional input-output tables specify foreign sources to domestic inputs by both country and sector of origin, e.g. such as in the original WIOD13 yearly data tables, the national input-output tables in the WIOD dataset discards the specific country origins of such foreign input, aggregating these foreign inputs by 34 'foreign' sectors.

The output from domestic sectors is either turned into intermediate domestic inputs within the Z matrix, being consumed or accumulated within the country (DFU), or exported abroad (E), the latter either for final use or as intermediate input to foreign sectors.

Input-output tables are used in many types of analyses, ranging from impact assessments of investments, demand mechanisms and spillover effects, to environmental analyses, calculating gross domestic products, and regional planning. Many of these analyses focus specifically on the inter-sectorial intermediate submatrices (Z and M), combining these with total sectorial outputs to determine technology coefficients (A) and the Leontief inverse (L), capturing the direct and total requirements to produce one unit of output per sector [see, e.g., 36]. When full information about the sectorial origins of imports is available, as in Table 1 below, the Z and M matrices are often combined (through addition) when calculating these coefficients [38]. Technology coefficients and the Leontief inverse thus indeed account for the sectorial origin of foreign intermediate inputs, albeit ignoring the specific country of origin of such inputs.

The complex valued directional network of intermediate flows within and between the economic sectors of an economy, where inputs originate both from within (Z) and outside (M) national boundaries, constitutes a core characteristic of the production structure of a national economy at a specific time. Whereas the direct (A) and total requirement (L) matrices have occasionally been described as the 'production structure' of a country, and compared as such [see, e.g., 38, also see 39–41], it is here argued that the de facto economic flows between and within sectors better represent a national production structure than what the conversion of such flows into the transformation functions of direct and total requirement coefficients do. The technology coefficients indeed capture the transformations that could happen in an economy, i.e. the amount and kind of input that is needed to create one unit of output, but that is

not necessarily reflective of the magnitudes of the transformations and activities that *do* happen in an economy.

Given the complexity of national production structures, where pairs of the Z and M matrices form a combined uni- and bipartite network (a so-called linked network; see [42]), what theoretically grounded computational approaches could be used for comparing such data structures on a pair-wise basis? Exploring previous comparative studies of national input-output structures, two broad analytical approaches can be observed.

## 2.1 Previous approaches for comparing national input-output networks

A classical approach for capturing characteristic and comparable features of a national production structure involves the triangulation of the inter-sectorial flows of a national input-output table [39, also 43]. By reordering sectors to maximize the sum of elements in the lower triangle of the intermediate flow matrix, the resulting optimal order of sectors arguably captures a hierarchy of production for a particular economy and year, an ordering which subsequently can be compared for those of other countries and years [e.g. 40, 41, 44, 45]. Several such studies have found high rank order similarities of sectors, also between assumedly different types of economies [e.g. 39, 45], supporting the notion of certain fundamental structures that economies seem to share [41].

Whereas these findings of fundamental structures are interesting, they are also indicative of a couple of general drawbacks with using triangulation for comparative purposes. First, triangulation assumes a linearity of production, in which potentially circular sectorial relations, albeit possibly rare [see 46], are ignored. Secondly, rank order correlations capture the ordinal sequence of items, while ignoring the finer details and continuous intervals underpinning such sequences [see also 40, 45]. Third: as reflected by the scarcity of triangulation studies using multiregional input-output data, it remains unclear how the triangulation approach could be adapted to the context of transnational production, i.e. in the presence of both domestic and foreign intermediate inputs.

Other studies take a more network-analytical approach to map and compare networks of national production. Many of these extract structural metrics at the nodal (sectorial) level, typically various kinds of centrality indices [e.g. 47–51] but there are also studies looking at other kinds of nodal and dyadic metrics [e.g. 52, 53]. Although not explicitly framed as such, the 1991 proposal by Dietzenbacher [31, also see 54, 55] does reflect a network-analytical approach to the study of national production structures. Building on Hirschman's ideas on unbalanced growth and the importance of industrial backward and forward linkages, Dietzenbacher suggested an eigenvector-based approach to capture such linkages. The dominant left-hand eigenvector (i.e. the eigenvector with the largest corresponding eigenvalue) of the direct requirement matrix (A) is here operationalizing the magnitude of sectorial backward linkages. Correspondingly, sectorial forward linkages are captured by the dominant right-hand eigenvector on the 'output matrix' (B), the latter containing the shares of one unit of output from a sector that flows as intermediate input to the other sectors. Using 1948–1984 data for the Netherlands, it is observed that the average sectorial rankings remained surprisingly static over the 1948–1984 period [31].

The above approaches indeed look 'under the hoods', providing means to map and compare the complex networks of inter-sectorial flows that occur within national economies. However, with the increasing shares of sectorial intermediate input that flow across national borders, where the domestic intermediate sectorial flows of the Z matrix are supplemented with the foreign intermediate inputs of the M matrix (see Table 1), the contours of national production structures are becoming increasingly blurry. Specifically, whereas imported intermediate input should be included when operationalizing sectorial backward/upstream

linkages, e.g. similar to how contemporary technology coefficients are determined, it can indeed be questioned (see subsequent section) whether a corresponding operationalization of sectorial forward/downstream linkages of a national economy also should include such intermediate flows *from* foreign sectors. Additionally, the choice between either comparing the networks of intra- and inter-sectorial transaction flows, or the technology- and output-coefficients derived from such transactions, is not an arbitrary choice, even though triangulation of either type of data might lead to the same hierarchy. Two economies sharing similar technology and output coefficients could indeed have very different economies in terms of inter-sectorial transactions.

Addressing the shortcomings of the comparative approaches above, the next section proposes and specifies a novel network-analytical framework for the mapping and comparison of national production structures in a transnational production regime.

## 2.2 Specifying the proposed fingerprinting approach

Inspired by Dietzenbacher [31], the approach proposed in this article uses directional left- and right-hand eigenvectors to extract a structural fingerprints of an economy, capturing the prominence of up- and downstream linkages for each economic sector in an economy for a specific year. However, the fingerprinting approach departs from Dietzenbacher's 1992 approach in two ways. First, instead of determining sectorial backward and forward linkages using the direct technology and output coefficient matrices, respectively, the fingerprint approach uses the sectorial flow networks to determine up- and downstream sectorial prominence. The aim here is thus not to capture and compare the coefficients of sectorial production functions of a specific country and year, but rather the specific intersectoral flow patterns that characterize its economic activities and production structure.

Second, whereas the proposed approach captures sectorial *upstream* prominence using the combined intermediate sectorial inputs from both domestic and foreign sources, i.e. the total intermediate input that flows into each of the domestic sectors, the *downstream* prominence metrics are determined based on domestic sectorial intermediate output alone, i.e. the total intermediate output that flows out from each of the domestic sectors. Specifically, as given in Eqs 1 and 2 below, the downstream prominence vector *d* is here operationalized as the dominant right-hand eigenvector of the Z matrix, whereas corresponding prominence of upstream flows *u* is captured by the dominant left-hand eigenvector of the summed Z and M matrices (T representing the element-wise summed together Z and M matrices).

$$Zd = \lambda_{max,d}d \qquad (Eq1)$$

$$uT = \lambda_{max,u}u \qquad (Eq2)$$

Underpinning the motivation for this specific operationalization, a visual representation of domestic (Z) and foreign (M) intermediate flows of a 4-sector toy economy is given in Fig 1 below. As the aim is to capture structural properties of a domestic production structure, a measure of upstream prominence of a domestic sector should preferably capture total intermediate input from other sectors, which in a transnational production regime involves intermediate input from both foreign and domestic sectors. As such, upstream sectorial prominence is determined in a similar way to how technology coefficients are calculated, i.e. by adding together both domestic (Z) and foreign (M) intermediate inputs. These are represented by the inbound black (Z) and gray (M) arrows in Fig 1 below.

To capture the corresponding downstream prominence of domestic sectors, we should similarly capture the total intermediate output from these sectors that end up in other parts of the

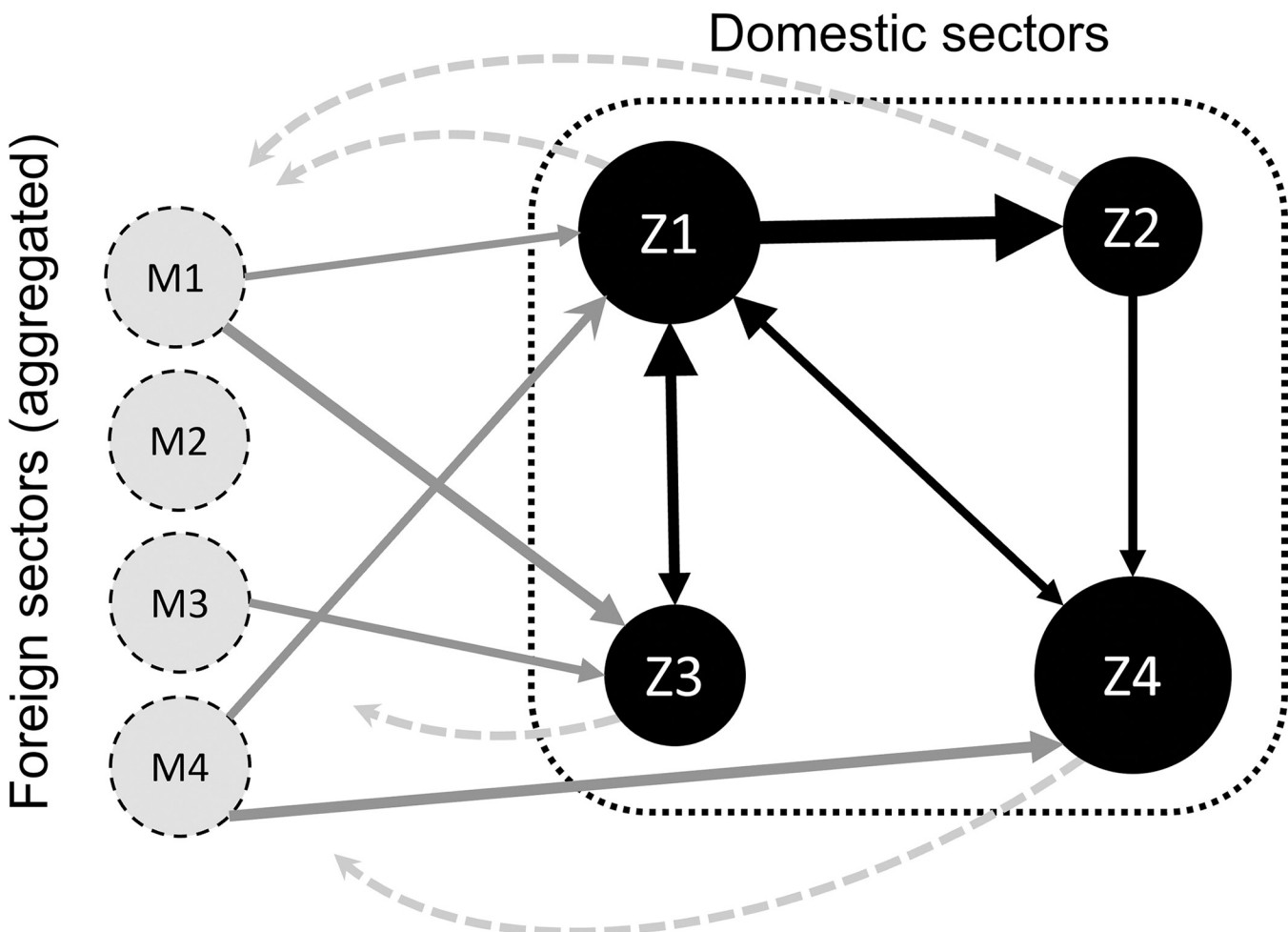

**Fig 1. A 4-sector toy example of a domestic production structure.** Nodes Z1-Z4 corresponds to four domestic sectors and nodes M1-M4 represents foreign sectors providing intermediate input to the domestic sectors. Note that the M sectors are aggregated by sector: e.g. M1 represents the 1$^{st}$ sector in all of the foreign countries.

domestic production structure. Corresponding to the outbound black arrows in Fig 1 below, the measures of sectorial downstream prominence are thus calculated using the Z matrix alone, i.e. similar to how the output coefficients (B) were calculated in [31]. If the *d* vectors instead were to be determined using the combined Z and M matrices (i.e. the T matrix), downstream sectorial prominence vector would then be based on *both* the domestic intermediate output to the domestic economy *and* the corresponding intermediate output of all foreign sectors to domestic sectors. Arguably, the specific downstream properties of all foreign sectors connected to a set of domestic sectors are not a constituent part of the properties of the domestic sectors.

What about domestic sectorial output that ends up outside the intermediate flows of the Z matrix? For sectorial output ending up as final use, domestic or foreign, such output is not accounted for in the downstream sectorial prominence vectors. As the aim is to capture structural properties of national *production* structures, such measures are not necessarily similar to profiles of total output, exports, or consumption. But what about sectorial output that ends up as intermediate input to foreign sectors? Such flows are instead accounted for by those various foreign structures of production, i.e. as contributions to their respective M matrices of foreign

intermediate goods. The fingerprinting approach is thus solely focused on the *production* structure of a *national* economy, i.e. the national components of transnational production networks, which by all means could be supplemented with additional descriptives of export composition, sectorial value-added, and structural positionality in the global networks of production and trade at large.

The fingerprint (f) for a specific country *c* and year *y* is subsequently obtained by concatenating the upstream (u) and downstream (d) sectorial vectors (where the ⊕ symbol represents the concatenation of two vectors):

$$f_{c,y} = u_{c,y} \oplus d_{c,y} \qquad\qquad \text{(Eq3)}$$

Exemplifying how a singular fingerprint can look like, Fig 2 below depicts the WIOD13-derived fingerprint of the production structure of Denmark in 1995. The top section depicts the upstream (input) vector *u*, and the bottom downward-pointing bars corresponds to the sectorial downstream prominence vector *d*. A notable feature of the Danish economy in 1995 is the prominent upstream linkages of its Food sector, while its downstream sectorial prominence is quite small. The output of this specific sector is indeed more likely to end up as final use than as intermediate input to other sectors, which implies that such end uses are not captured by the fingerprinting approach. The up- and downstream prominence of the Business and Finance sectors are somewhat reversed, where the prominence of their domestic downstream linkages reflect their common role of serving other sectors of the domestic economy.

How does this Danish example compare with the average structural fingerprint of all country-year fingerprints in the WIOD13 dataset? The solid dots in Fig 3 below represent the average up- and downstream sectorial prominences for 40 countries in the period 1995–2011, with lighter bars capturing spread (as one standard deviation from the means). An ocular comparison with the Danish fingerprint for 1995 hints at a more pronounced upstream prominence

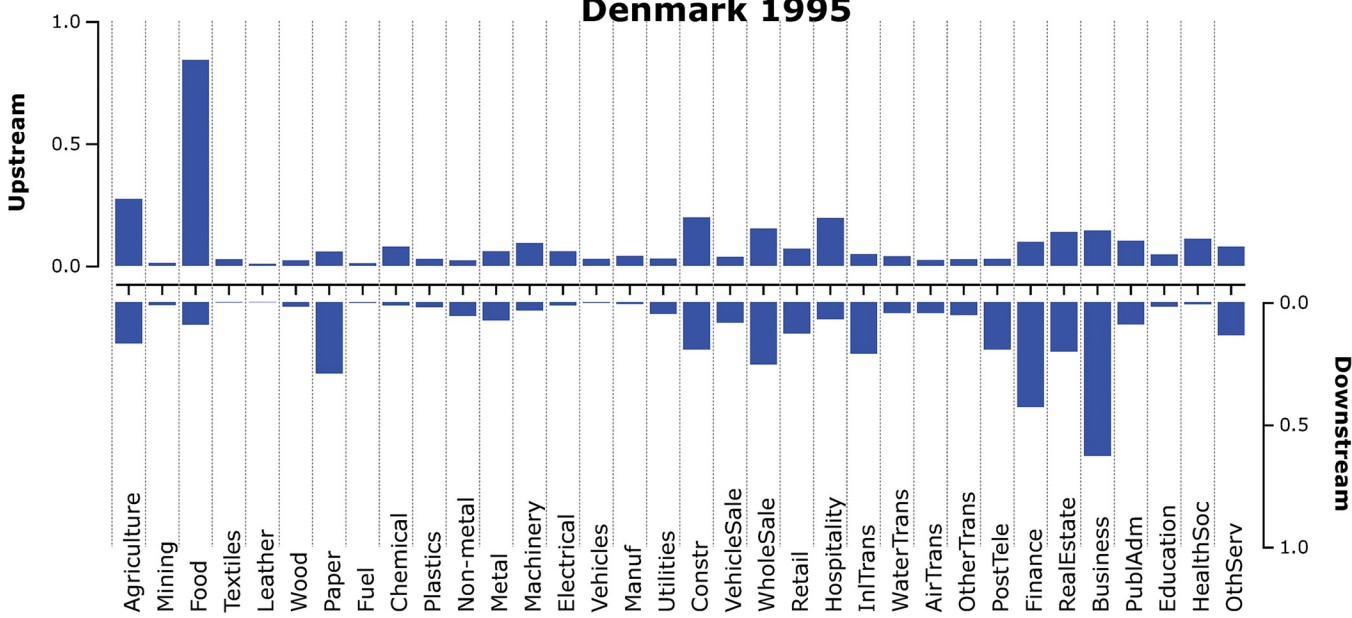

**Fig 2. Fingerprint of the Danish production structure in 1995 [D3js].** Upstream sectorial prominence are given by the upward-directed top bars, and downstream sectorial prominence are given by the downward-directed bottom bars. See Appendix A.2 for sector details and the WIOD mapping to ISIC rev.3. Eigenvalue diagnostics(DNK_1995): $u_\% = 27\%$, $u_{1vs2} = 1.38$, $d_\% = 36\%$, $d_{1vs2} = 1.45$ (see Section 2.3).

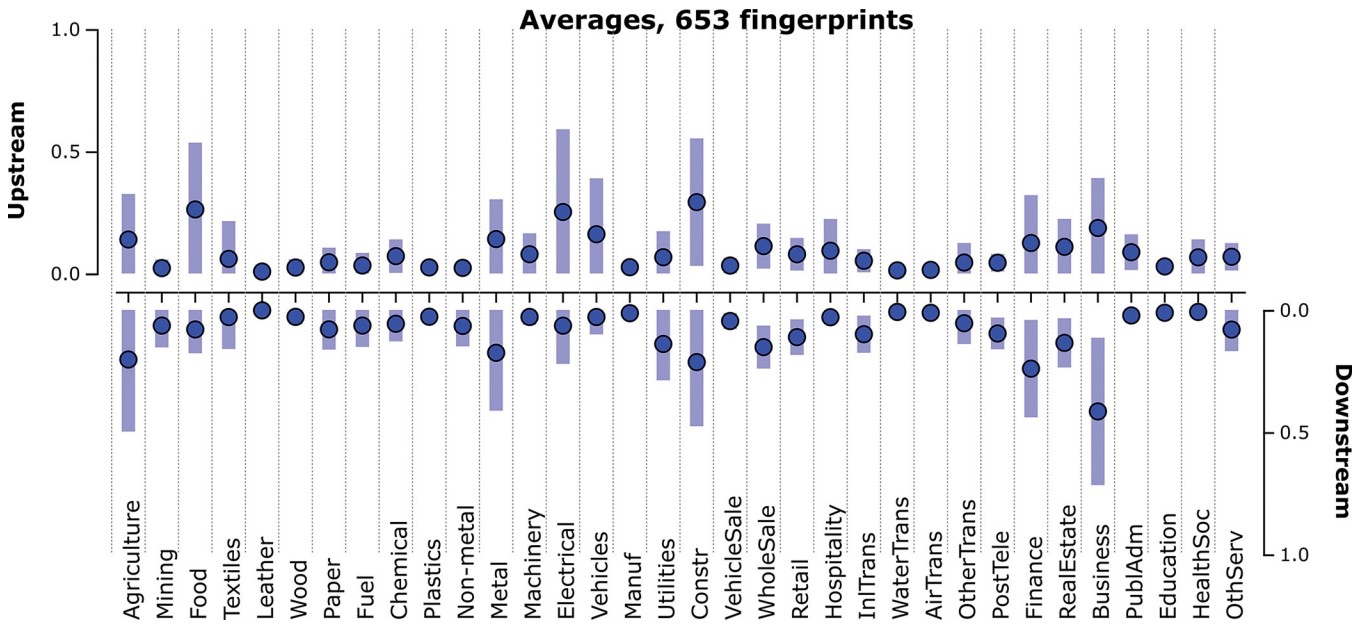

**Fig 3. Average fingerprint of 653 production structures for 40 countries in the 1995–2011 period [D3js].** Dots indicate average sectorial up- and downstream prominence, and the bars indicate one standard deviation above and below these average values. Note that 27 country-year fingerprints were removed (see Section 2.3).

for the Danish Food sector and a somewhat higher downstream prominence for its Business, Finance, and Paper sectors, compared to the average fingerprint.

How does the Danish production structure for 1995 differ from its north-eastern neighbor Sweden in the same year? To exemplify a pair-wise comparison between fingerprints, Fig 4

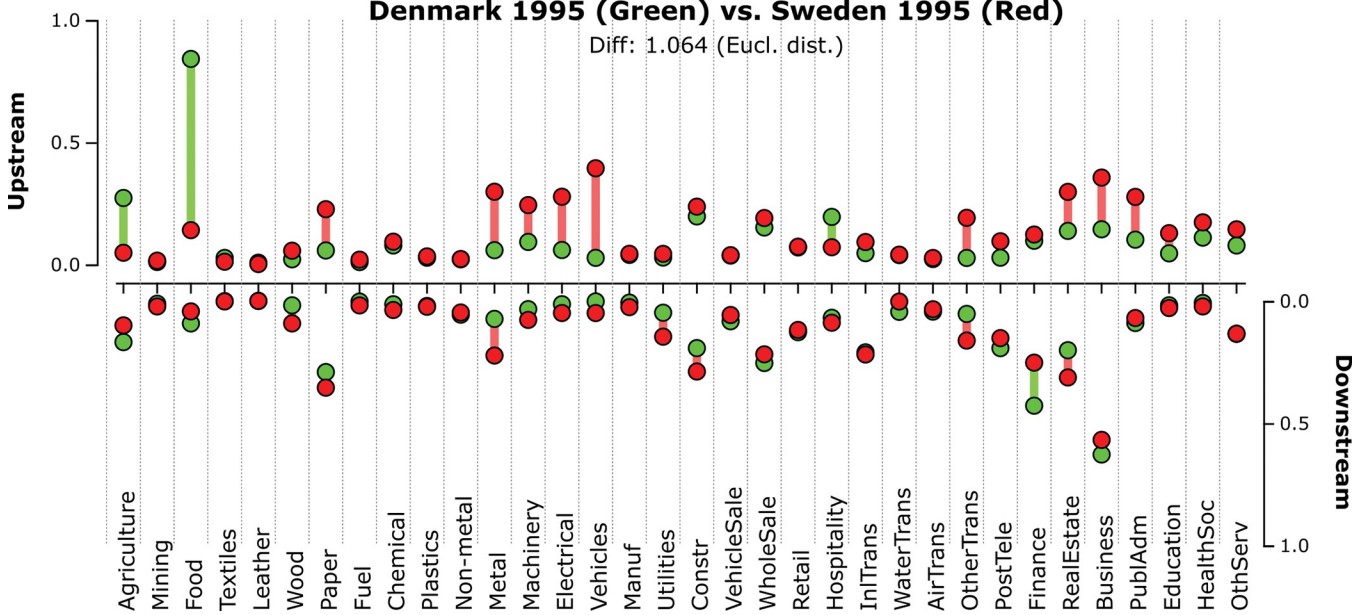

**Fig 4. Comparing fingerprints: Denmark 1995 vs. Sweden 1995 [D3js].** Green dots represent up- and downstream sectorial prominence for Denmark 1995, with red dots representing up- and downstream sectorial prominence for Sweden 1995. Colored bars indicate the difference in sectorial prominence between Denmark 1995 and Sweden 1995, where the color indicates whether it is Denmark (green) or Sweden (red) that has a higher prominence for that particular sector. Eigenvalue diagnostics(SWE_1995): $u_\% = 23\%$, $u_{1vs2} = 1.56$, $d_\% = 28\%$, $d_{1vs2} = 2.12$.

below depicts the fingerprint of both these production structures–Denmark 1995 (green dots) and Sweden 1995 (red dots)–with the colored bars indicating the relative differences between these two economies. The upstream prominence of the Danish Food sector is noticeably larger than that of its Swedish counterpart, whereas the upstream prominences of the Swedish manufacturing sectors are somewhat larger than in Denmark for this year.

With each fingerprint $f_{c,y}$ corresponding to a point in a 68-dimensional hypercube, the proposed measure of dissimilarity between two fingerprints is the Euclidean distance between two such points–see Eq 4 below.

$$dist(f_{c1,y1}, f_{c2,y2}) = \sqrt{\sum_{s=1}^{34} \left( u_{c1,y1,s} - u_{c2,y2,s} \right)^2 + \sum_{s=1}^{34} \left( d_{c1,y1,s} - d_{c2,y2,s} \right)^2} \qquad \text{(Eq4)}$$

(where $f_{c,y}$ is the fingerprint of country $c$ at year $y$, $u_{c,y,s}$ is the upstream prominence of sector $s$ for country $c$ at year $y$, and $d_{c,y,s}$ is the corresponding downstream prominence)

In the case of the two Nordic economies in Fig 4 above, the dissimilarity between the Swedish and Danish structural fingerprints in 1995 is 1.06. This is lower than the average pairwise dissimilarity measure for all 741 country-pairs in 1995 (1.41), yet far from as similar as what is the case for the Romanian and Lithuanian fingerprints of 1995 (0.15). For the complete set of 212,878 pairwise dissimilarity measures for all included countries and years in the WIOD13 dataset, the left-skewed distribution has a mean and median of 1.43 and 1.49 respectively (see Fig 5).

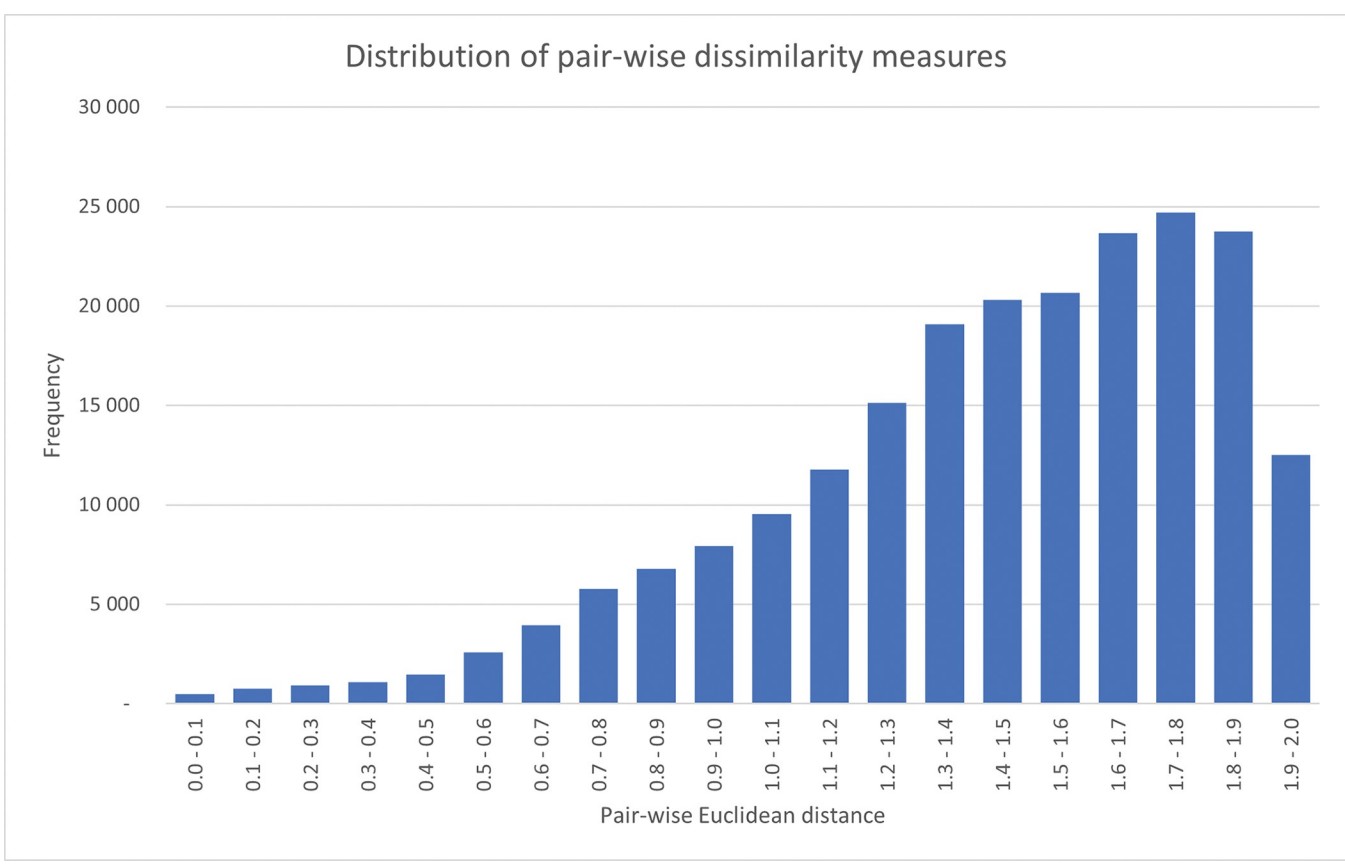

**Fig 5. Distribution of all pairwise Euclidean dissimilarities for the whole set of 653 fingerprints.** With a total of 212,878 pairwise country-year comparisons, the frequency plot is binned into 20 segments, each with a width of 0.1.

**Table 2. Statistics on the eigenvalue diagnostics derived from the 653 WIOD13 fingerprints.**

| | Upstream vectors (Z+M) | | Downstream vectors (Z) | |
|---|---|---|---|---|
| | 1st share ($u_\%$) | 1st/2nd ratio ($u_{1vs2}$) | 1st share ($d_\%$) | 1st/2nd ratio ($d_{1vs2}$) |
| **Mean** | 0.27 | 2.05 | 0.29 | 2.04 |
| **Median** | 0.24 | 1.54 | 0.27 | 1.66 |
| **Range** | 0.16–0.87 | 1.08–18.81 | 0.18–0.89 | 1.11–19.24 |

## 2.3 The eigenvalues of WIOD13-based fingerprints

A common method to evaluate how well extracted eigenvectors manage to capture latent features of more complex data involves examining the eigenvalues corresponding to these eigenvectors. As each fingerprint consists of two dominant eigenvectors derived from slightly different matrices (T vs. Z), each of these have their own set of decreasing eigenvalues. The size of these dominant eigenvalues with respect of remaining eigenvalues could thus provide insight on how well the complexity of these networks can be expressed in terms of these measures of sectorial prominence.

Two diagnostic measures were calculated for each dominant eigenvector. The first measure ($u_\%$ and $d_\%$) captures the size of the dominant eigenvalue as a percentage of the sum of all eigenvalues, and the second measure ($u_{1vs2}$ and $d_{1vs2}$) captures the ratio between the dominant and second-largest eigenvalue. A statistical summary of these four eigenvalue diagnostics for all 680 fingerprints in the WIOD13 database is given in Table 2 below (see Appendix D.1 for a full specification of the eigenvalue diagnostics).

Whereas the first eigenvalues on average are more than twice the size of the second eigenvalues, on average corresponding to 0.27 and 0.29 of the sum of all eigenvalues, several fingerprints do have eigenvectors whose eigenvalue diagnostics preferably would be larger. Few, however, have low values for *both* of their eigenvalue diagnostics. For the 27 fingerprints (see Appendix D.1) that do have low values for both the eigenvalue diagnostics ($u_\% < 0.2$ and $u_{1vs2} < 1.25$, or $d_\% < 0.2$ and $d_{1vs2} < 1.25$), either for one or both of their two eigenvectors, these were excluded.

Still, whereas one should be careful when interpreting fingerprints with relatively poor eigenvalue diagnostics, the dominant eigenvectors do reveal a significant structural consistency over time, even when keeping the above 27 fingerprints in the analyses. Thus, even though occurrences of low eigenvalue diagnostics could reflect structural dynamics that are not properly caught by the dominant eigenvectors, such dips do not seem to be associated with spurious transformations and transitions between types.

## 3. Case study 1: Towards a taxonomy of national production structures

To what extent can we identify different types of national production structures in the contemporary world economy of transnational production? Can we derive analytically useful classifications of nations, and potentially a taxonomy of such structures, based on similarities of their fingerprints of production structures? Which countries transition between different types over time, and which remain the same?

Addressing these questions, this case study applies exploratory cluster analysis on the full set of pairwise dissimilarity measures between the 653 country-year fingerprints derived from the WIOD13 dataset. Informed by suitable cluster adequacy metrics, an 11-cluster partition is proposed as an analytically useful partition, also exploring the partitions at the 5- and 21-cluster levels. By tracing how national economies transition between these 11 structural types, it

can be observed that several of these structural shifts coincide with specific economic-historical events at the national level.

How do these structural similarity patterns between countries and years relate to corresponding similarities of gross exports? Using the sectorial gross export profiles for each country and year, pairwise export dissimilarity measures were calculated, followed by a similar agglomerative cluster analysis as in the fingerprinting case. Comparing these gross-export-based clusters with those determined on the basis of the internal networks of production, it is argued that the latter are analytically more useful than the rather trivial clusters based on gross export similarities.

## 3.1 A 11-type classification of national production structures

Given a matrix of dissimilarities, such as the Euclidean distance measures of pairwise structural dissimilarity between the 653 country-year fingerprints, agglomerative hierarchical clustering is a useful and intuitive technique for exploratory analysis [56]. In this approach, the two items that are the most similar (nearest distance) are first identified, subsequently merging these into a joint item (cluster), followed by recalculating all distances between this newly merged cluster item and all remaining items in the matrix. The procedure is then repeated, finding and merging the next two items/clusters that are the most similar, until all items have been merged into a singular cluster. This yields a hierarchy of nested partitions, from which a suitable partition can be chosen, the latter which is typically informed by various cluster adequacy metrics.

When two items and/or clusters are merged, the subsequent recalculation of between-item distances can be done in different ways. In so-called 'complete-link' clustering, new distances are determined based on the largest distances between the individual items in two clusters, whereas in 'single-link' clustering, new distances are determined on the smallest dissimilarity between the individual items. Complete- and single-link clustering being only two out of several potential methods for agglomerative clustering, the choice of such a method for recalculating distances typically have a notable impact on the cluster hierarchy that is obtained.

In this exploratory cluster analysis of structural fingerprints, the so-called unweighted average-link (UPGMA) clustering method was chosen. With this clustering method, the distance between two clusters is determined as the unweighted average of all pairwise distances between items in respective cluster. Contrary to several other clustering methods (such as the weighted average-link and Ward methods), the UPGMA method has no inherent bias for partitions where clusters are of similar sizes. Contrary to complete-link clustering that has a preference for low within-cluster distances, UPGMA has an intuitive balance between low within-cluster dispersal and high between-cluster dispersal. That said, despite perhaps being the geometrically most intuitive agglomerative clustering approach, making it useful for these kinds of initial exploratory analyses, this does not rule out arguments in favor of other clustering techniques.

Whereas agglomerative hierarchical clustering provides a full hierarchical set of potential partitions, which in this case is 653 different partitions (i.e. including the trivial partitions where all fingerprints either are in the same cluster, or each fingerprint represents a unique singleton cluster), the choosing of suitable partitions is left to the analyst. To guide such choices, several types of cluster adequacy metrics exist. Reflecting a balance between good separation between reasonably tight clusters, the Calinski-Harabasz (CH) cluster adequacy metric captures the ratio between between-cluster and within-cluster dispersal. A distinct peak in this metric for a particular partition, i.e. where the CH metric of surrounding partitions are notably lower, could thus indicate a particularly suitable partition.

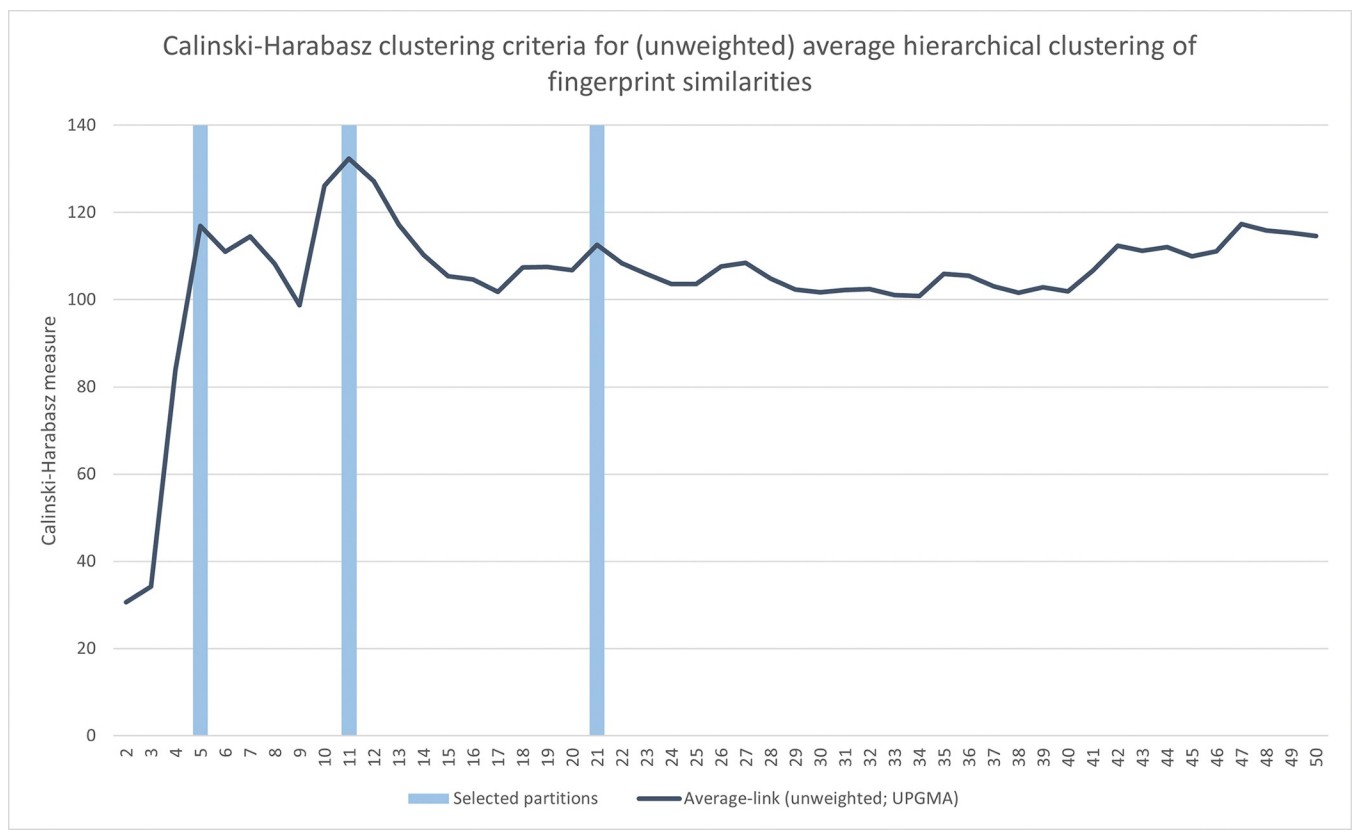

**Fig 6. Cluster adequacy analysis for the (unweighted) average-link hierarchical clustering of 653 fingerprints.** The plot captures the Calinski-Harabazs clustering criteria when checking all clusters in the 2–50 range emerging when applying unweighted average hierarchical clustering on pairwise country-year fingerprint Euclidean distances.

Using the full dissimilarity matrix of 653 country-year fingerprints, applying unweighted average-link agglomerative clustering to this matrix, Fig 6 below captures the Calinski-Harabasz metric for obtained partitions in the range of 2–50 clusters. As there is a notable peak around the 11-cluster partition, this was chosen as a potential classification candidate to explore further. Two additional partitions were also included as reference, at the 5- and 21-cluster partitions respectively. Looking at the mean and spread of pair-wise between- and within-cluster distances for these three partitions (Fig 7 below), cluster separation seems to improve from the 5- to the 11-cluster partitions, especially with respect to within-cluster dispersal, where the subsequent improvement at the 21-cluster partition is seemingly more marginal.

A sunburst chart for the 5-, 11- and 21-cluster partitions is presented in Fig 8 below. The cluster labels are here based on an interpretation of the characteristic average fingerprints of each cluster, particularly those patterns that emerge at the 11-cluster partition.

The 5-cluster partition is dominated by a cluster containing more than half of all country-year fingerprints and with most countries represented (27 of 40), all sharing sectorial profiles centered around Business, Construction & Vehicles (see Fig 9A). Notably, this cluster contains the full temporal sets for several West- and South-European countries, with Russia, China, India and several Central- and East-European countries as intermittent cluster members. At the subsequent 6- and 7-cluster partitions, Vehicles and Metal & Fuel form separate clusters, further consolidating the West- and South-European character of the remaining countries.

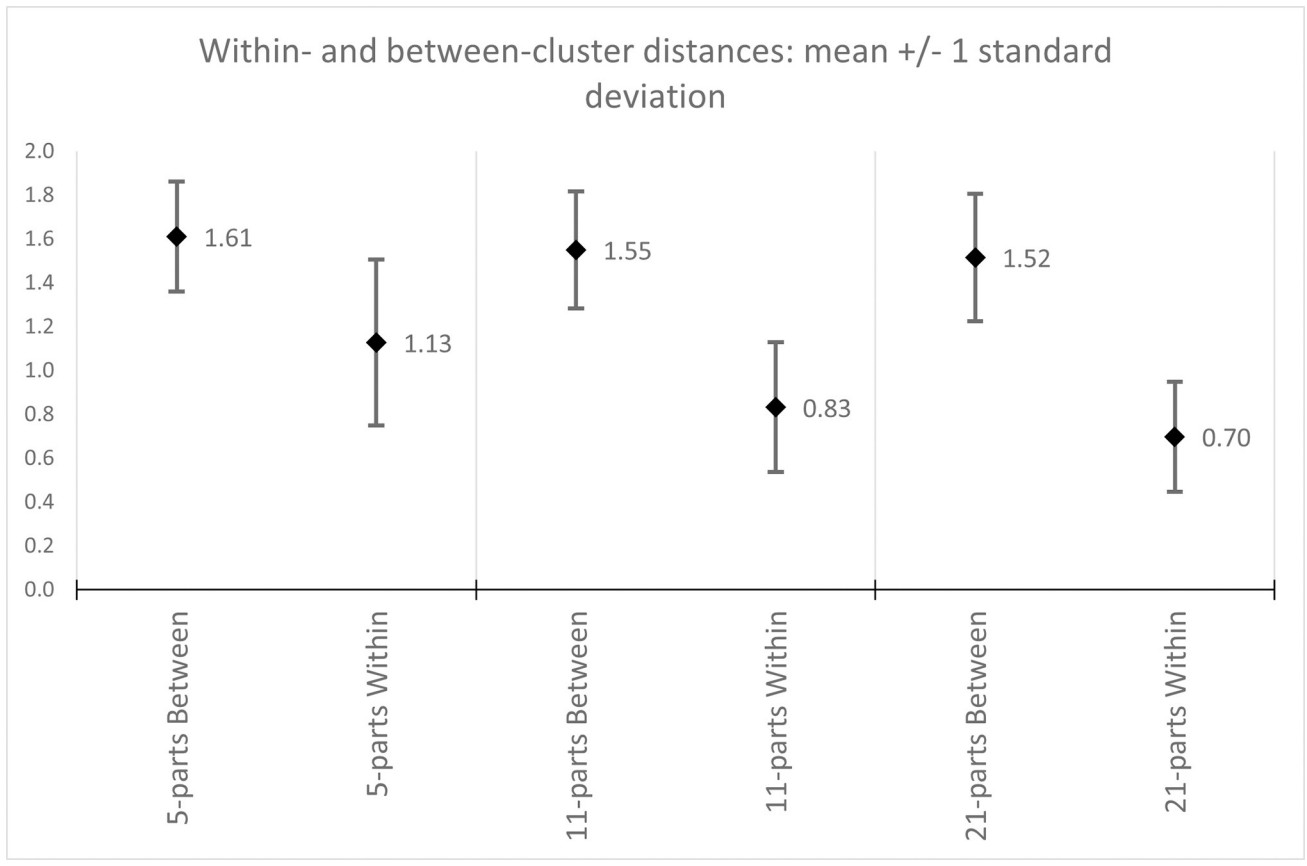

**Fig 7. Within- and between-cluster distances, 5-, 11-, and 21-cluster partition from (unweighted) average-link clustering of 653 fingerprints.**
Within-cluster distances constitute all pairwise distances of country-year fingerprints that are clustered together, and between-cluster distances constitute the remaining pairwise distances of country-year fingerprints that are not clustered together. These are represented by both the average distances as well as one standard deviation above/below these averages to capture the spread of values.

This large cluster is followed by two clusters characterized by Electrical & Metal (Fig 9B), and Agriculture & Food (Fig 9C), followed by two smaller clusters capturing distinct specializations in respectively Textiles and Finance.

Whereas the size of the Business, Construction & Vehicles cluster might undermine the analytical usefulness of this partition, it is interesting that these production structures are this similar to each other. In terms of domestic production structures, there thus seems to be some support for a 'European economy' moniker at this level–and also a corresponding 'Western developed economy' moniker at the 11- and 21-cluster partitions.

Similar to the smaller specialized Textile- and Finance-oriented clusters, the Agriculture & Food subset remains the same at the 5- and 11-cluster partitions, but the other two major subsets of the 5-cluster partition split into more specialized types at the 11-cluster partition. Business and Construction emerge as two distinct types, broadly separating the business-oriented West-European countries from its more construction-focused southern and eastern neighbors. Notable exceptions are Latvia (1998–2001) and Estonia (2007–2011), both which form their own joint cluster at the subsequent (12-cluster) partition. Japan and Canada, together with late-period Slovakia, form a Vehicles-oriented type, a Metal & Fuel-oriented cluster with Russia (from 2003), India (from 2004), and China (1995–1999) emerges, and Austria (2009–2011) and Slovakia (2001–2003) form a small peculiar 'Utilities'-specialized cluster.

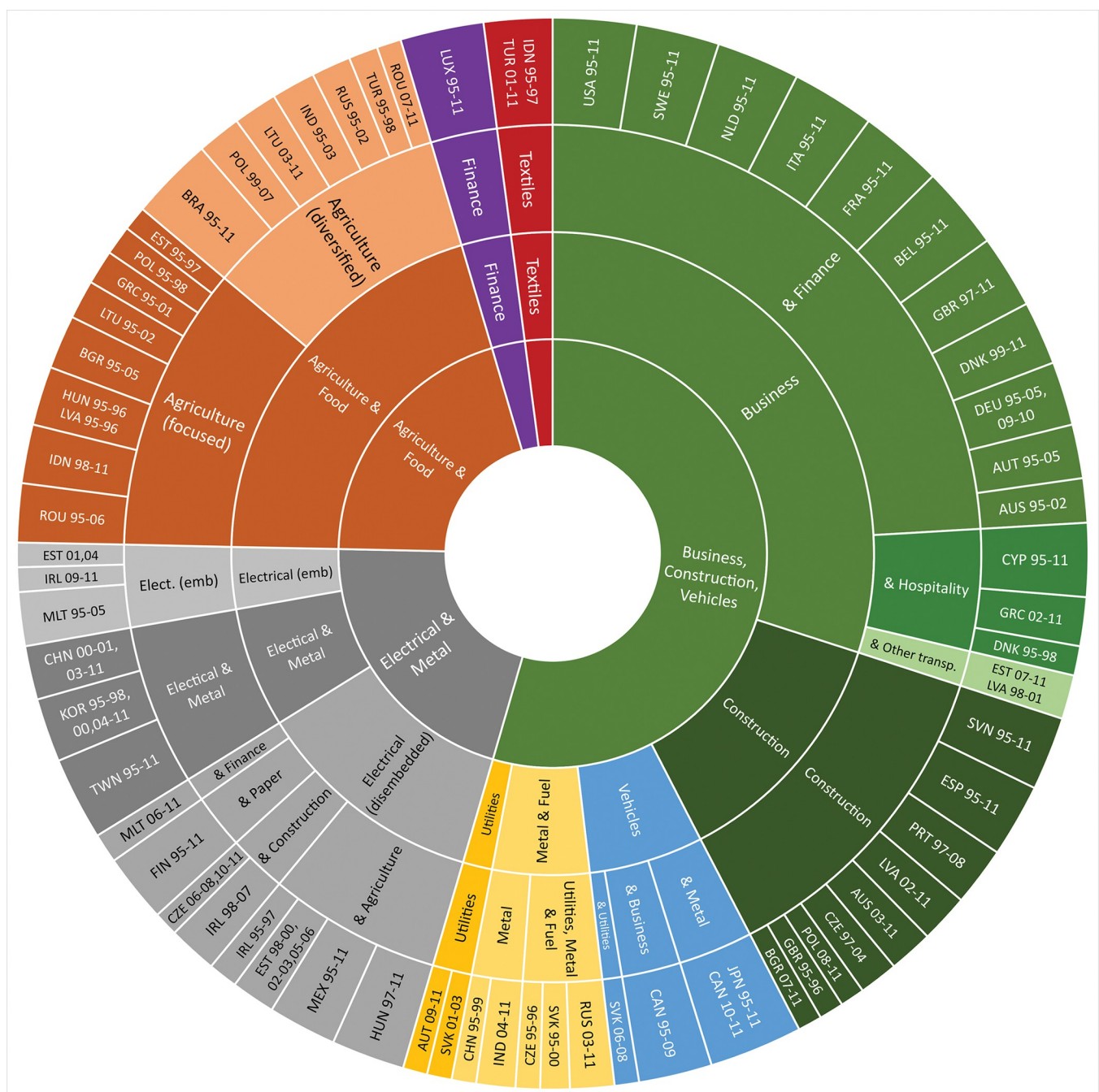

**Fig 8. Sunburst chart of the 5-, 11- and 21-cluster partitions from (unweighted) average-link clustering of 653 fingerprints.** Countries and years are grouped together on the basis of similarities of their fingerprints. As the number of clusters increase, country-years within clusters are more similar. Note that Turkey 1999 and Bulgaria 2006 are here excluded, both constituting singleton clusters in the 21-cluster partition.

The Electrical & Metal-oriented fingerprints separate into three subsets at the 11-cluster partition, all sharing prominent Electrical upstream flows but with different secondary specializations. Taiwan, South Korea, and China (2000–2011) all have domestically well-embedded Metal sectors, which the other Electrical-oriented sibling economies lack. A broader set of Electrical-oriented countries, all with fairly undeveloped Electrical downstream ties to their other sectors, have secondary specializations that are more visible at the 21-cluster partition.

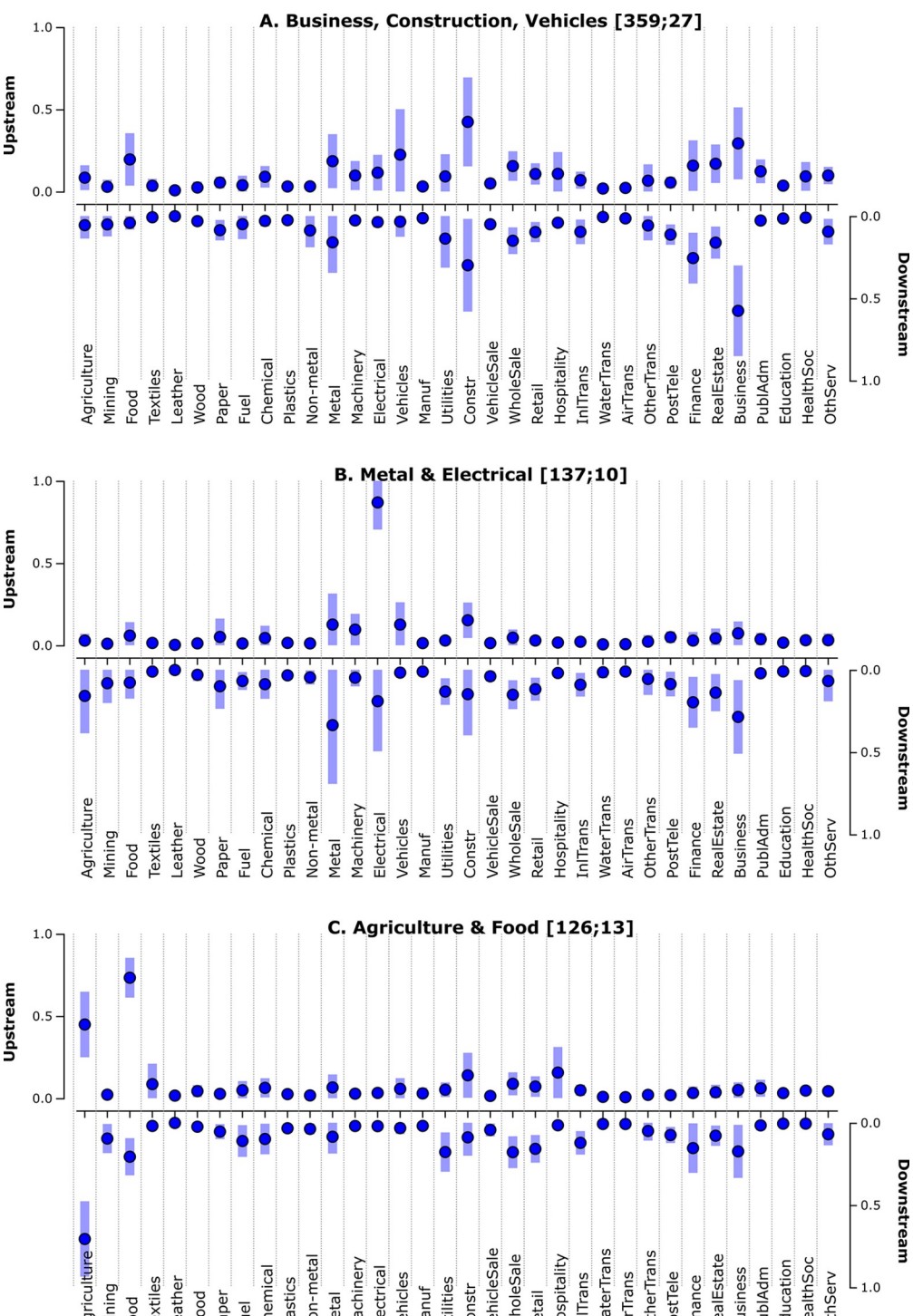

**Fig 9. Average fingerprints (mean and +/- 1 SD) for major clusters in the 5-cluster partition [D3js].** Dots indicate average sectorial up- and downstream prominence, and the bars indicate one standard deviation above and below these average values.

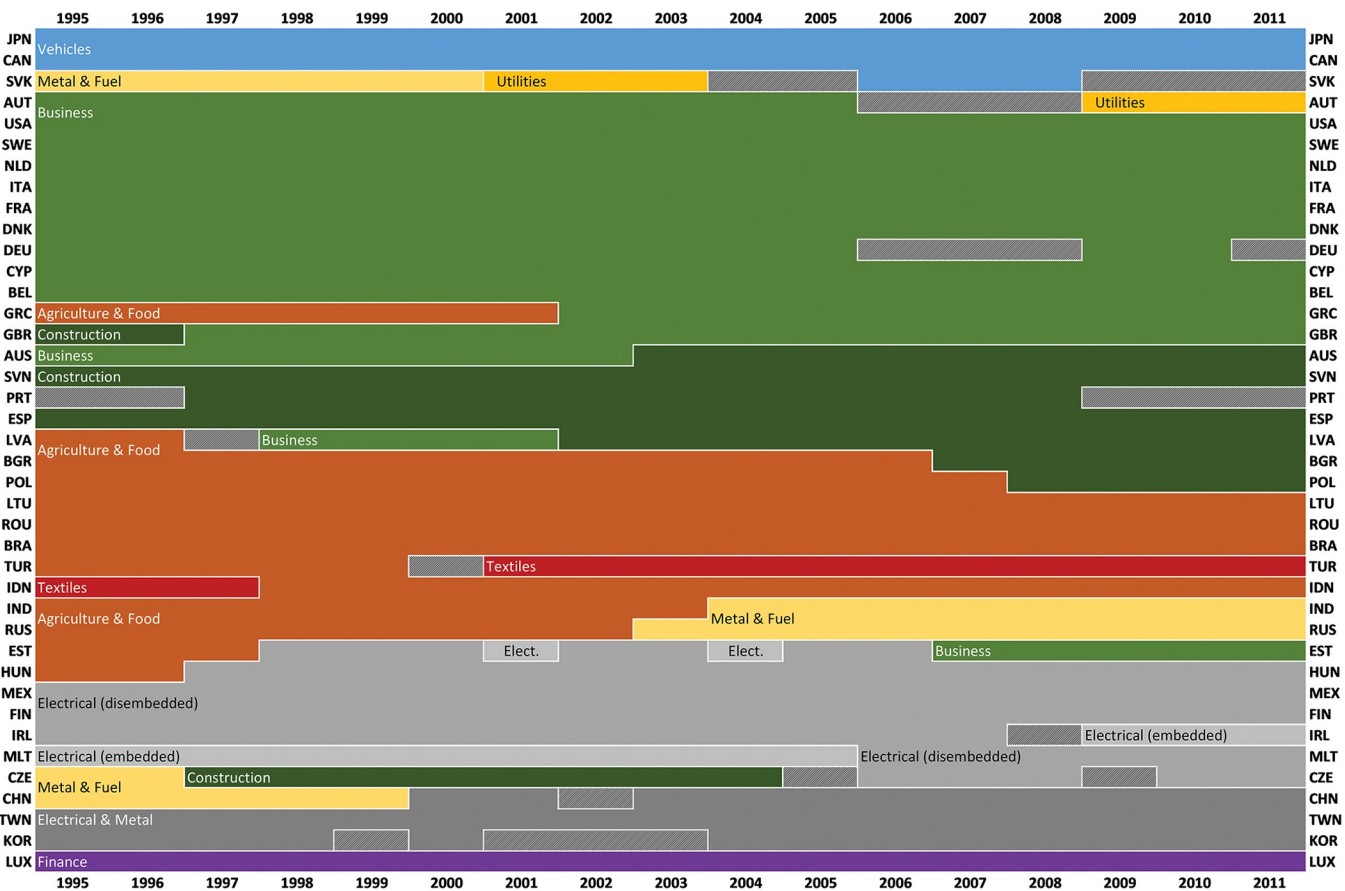

**Fig 10. Longitudinal transformations of national production structures: a sequence index plot of the 11-cluster partition.** Each color represents one of the 11 clusters obtained using (unweighted) average hierarchical clustering (see Fig 8).The 27 country-year fingerprints without colors were excluded due to their poor eigenvalue diagnostics; see section 2.3.

However, a smaller and fairly distinct cluster with Malta (1995–2005), end-period Ireland, and two Estonian fingerprints all have Electrical sectors that are notably well-embedded in their domestic production structures, which indeed differs from other economies with this sectorial specialization.

Providing an overview of longitudinal change and transitions between structural types, the sequence index plot in Fig 10 below represents an alternative way to visualize the 11-cluster partition. Of the 12 countries classified as Agriculture & Food-oriented in 1995, only Lithuania, Romania, Brazil, and Indonesia remained so in 2011. Contrasting this, most countries of the Business type remained so during the whole 1995–2011 period.

Several country-specific transitions between types seemingly coincide with specific economic-historical events. Greece's 2002 transition to the Business-type coincides with its adoption of the Euro; Malta's 2006 shift to a domestically dis-embedded electronics industry coincides with both a general deceleration of industrial value added growth rates [57], economic policy adjustments following its 2004 EU membership, and a major restructuring of Malta's main electronics industry [58]; and Russia's structural transformation from Agriculture & Food to Metal & Fuel in 2003 coincides with a drastic increase in international oil prices [e.g. 59]. Although highly selective and not conclusive in any way, these observations indicate that this partition could function as an analytically useful classification of national production structures.

At the 21-cluster partition, additional distinctions emerge that seem to reinforce the analytical usefulness of the fingerprinting approach. Cyprus, Greece, and early-period Denmark form a hospitality-oriented sub-type in the Business cluster, Agriculture & Food is separated into, respectively, a focused and a diversified cluster, and the Vehicles and Metal & Fuel types from the 11-cluster partitions both break up into more specialized types at the 21-cluster level. For the economies with domestically dis-embedded Electrical sectors, different types emerge with different secondary specialization: Paper for Finland, Finance for late-period Malta, and Hungary indeed seems to be a "Mexico of the East" [60], sharing prominent Agricultural sectors with early-period Ireland (1995–1997) and intermittently also Estonia.

## 3.2 Alternative clusters: comparing structural fingerprinting with gross export similarities

If we instead turn our attention to the cross-border exports of national economies, comparing the sectorial composition of gross exports of countries and years, how do such similarity patterns compare with the corresponding patterns of domestic production structure? Using the gross export vectors E for all countries and years in the WIOD13 dataset, i.e. here including the 27 countries and years that were excluded from the previous analysis, these export vectors are first marginal-normalized and subsequently compared on a pairwise basis by determining the Euclidean distance between these normalized vectors. As with the fingerprinting clustering above, these measures of gross export dissimilarities are subsequently used as input to unweighted-average agglomerative clustering, using the Calinski-Harabazs cluster adequacy to identify a suitable number of partitions.

Contrary to the fingerprint clustering above, the cluster adequacy test produced more ambiguous results for the analysis of gross sectorial exports. As shown in Fig 11 below, the Calinski-Harabazs index continues to increase as the number of partitions increase. However, as there is a notable increase in the adequacy metric when going from 15 to 16 clusters, this being reasonably close to the number of clusters in the proposed classification above, the 16-cluster was here chosen, shown as a sequence index plot in Fig 12 below.

Compared with the 11-cluster partition based on the proposed fingerprinting approach (Figs 8 and 10), the analysis of gross sectorial export similarities yields clusters that are quite different. Gross sectorial export profiles are to a large extent both static and country-specific: 10 out of the 16 clusters contain all years for individual countries. Transitions are also somewhat rare: only four countries shift from one type to another during the 1995–2011 period, seemingly not capturing the same kind of historical dynamics that is observed when looking at the internal structures of production. Although the shared specializations in electrical sector industries can be observed, where two clusters emerge, there is a notably lack of many other expected distinctions and trade specializations, such as Agriculture & Food, Vehicles, Construction, Textiles etc.

The observed differences between fingerprinting and gross export clustering results do not in any way invalidate analytical approaches and classifications based on gross sectorial exports, which indeed shed light on how national economies have different specializations in the international exchange of both intermediate and final goods. What these differences do show, however, is that the proposed fingerprinting approach provides insights on the internal structures and dynamics of national production which more output-oriented approaches might miss.

## 4. Case study: regional structural transformations during the Eastern expansion of the European Union

This second case study addresses the Central-Eastern enlargement of the European Union [33, 61–64], which has been viewed as a natural experiment [65, 66] for testing theories of market

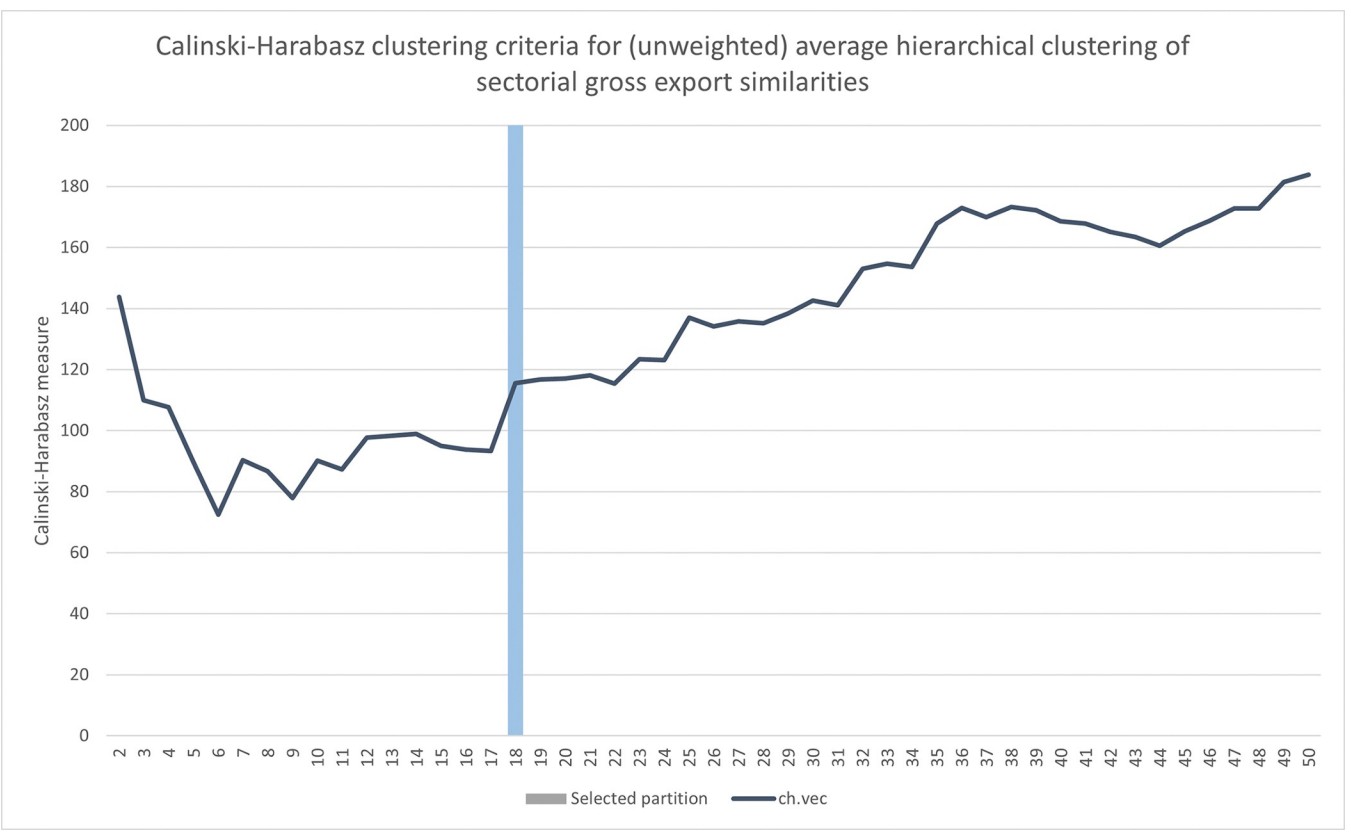

**Fig 11. Cluster adequacy analysis for the (unweighted) average-link hierarchical clustering of 653 sectorial gross export vectors.** The plot captures the Calinski-Harabazs clustering criteria when checking all clusters in the 2–50 range emerging when applying unweighted average hierarchical clustering on pairwise country-year sectorial gross export Euclidean distances.

integration and trade [67–70]. Specifically, this case study tracks the production-structural trajectories of the set of 'Eastern' countries that joined the EU in 2004 –the so-called 'A8' countries consisting of Czech Republic, Estonia, Hungary, Latvia, Lithuania, Poland, Slovakia, and Slovenia–and also Bulgaria and Romania, both joining three years later. Contrasting these, the West-European region is represented by five of the six founding members of the EU: Belgium, France, Germany, Italy, and the Netherlands, labeled here as the 'Core 5' group. (Due to its seemingly unique and stable production structure, Luxembourg was excluded here. If included, the average structural transformation for the Western set would be even lower). Removing the 12 country-year fingerprints with low eigenvalue diagnostics, a total of 243 country-year fingerprints is covered here.

Viewed through the lens of structural fingerprinting: how did the structural transformation of national economies look like in the East and the West respectively in the period 1995–2011? Did these transformations result in a diversification of national production structures within each region? Of particular relevance for theories on integration and economic specialization: did the national production structures of respective region converge or diverge during this period?

Fig 13 captures the cumulative annual structural change in each country from 1995 to 2011. With the exception of Slovenia and Germany, the Eastern economies experienced more aggregate structural transformation than their Western counterparts during this period. Confirming what was noted in the previous case study (see Fig 10), the fingerprinting approach suggests

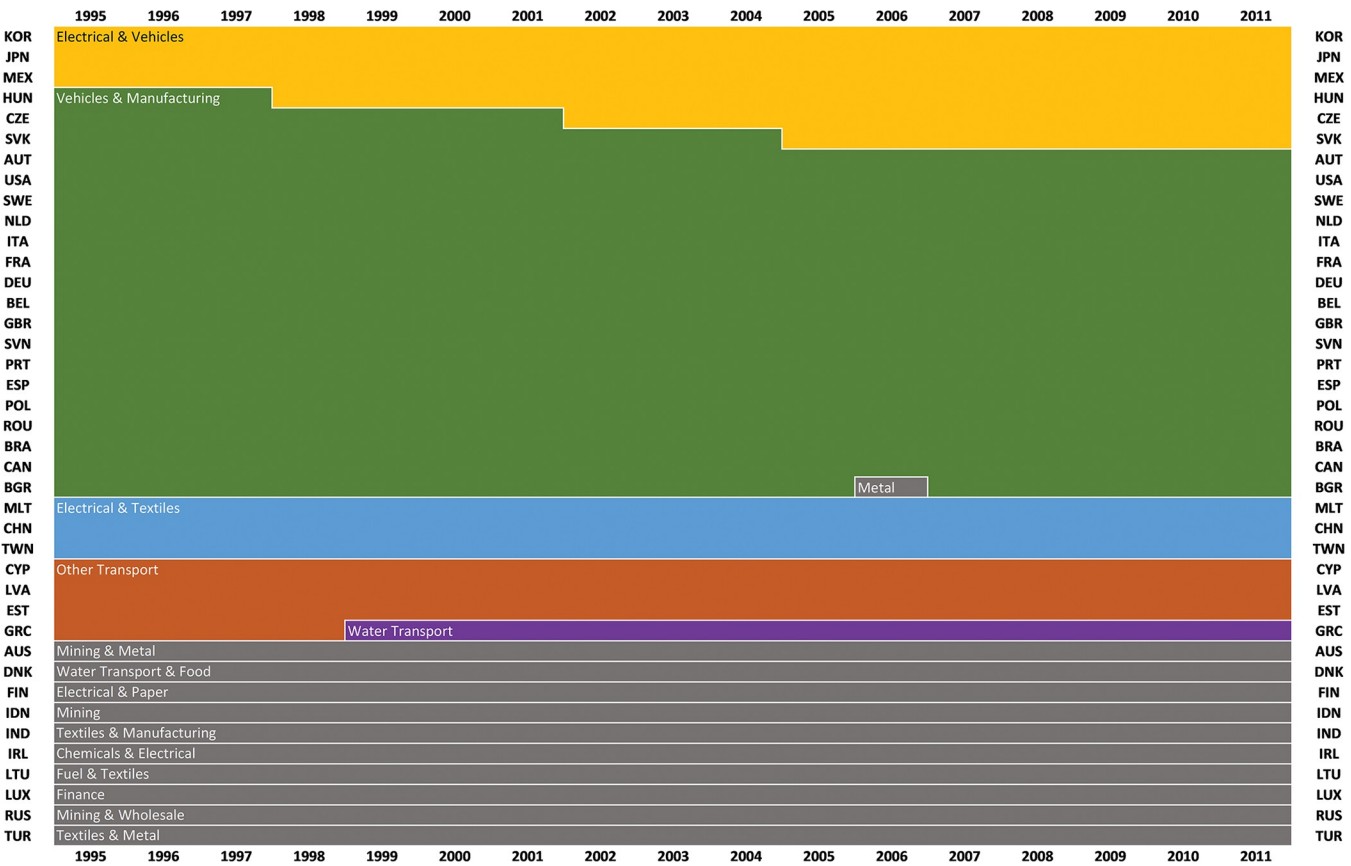

**Fig 12. Sequence index plot: 16-cluster partition based on gross sectorial export similarities.** Each color represents one of the 11 clusters obtained using (unweighted) average hierarchical clustering of gross export Euclidean distances.

that it is specifically Estonia's production structure that transforms the most during this period. The dramatic transformations of the Eastern economies seem unrelated to their year of accession; and indeed, this 'great transformation' of the Central-East European economies began long before their formal accession to the European Union [71].

What did these annual transformations imply for the overall trajectories of the national production structures? By setting 1995 as the index year to which we compare subsequent annual fingerprints, Fig 14 below captures a similar story: whereas the Western production structures in 2011 were not that different from their 1995 versions, the Eastern production structures anno 2011 were indeed quite different from their historical counterparts. Similar trajectories emerge when indexing from 2004 (see Fig 15), where the Eastern production structures (and particularly those of Estonia and Bulgaria) anno 2011 being quite different from their 2004 counterparts, while the Western structures on average remained relatively similar. The continuous transformations of the Eastern national production structures that occurred over the period 1995–2011 were thus not merely oscillations around country-specific structural types but represented more fundamental movements along new structural trajectories.

What did these national transformations mean for the structural diversity of respective regions? We operationalize the latter by tracking the pairwise structural dissimilarities for the countries within each region over time. Among the A8 countries, the relatively high mean dissimilarity in 1995 (1.10) increased even further (1.35)–see green solid line in Fig 16. It is noteworthy that the internal pairwise dissimilarity variance decreased at the same time (green

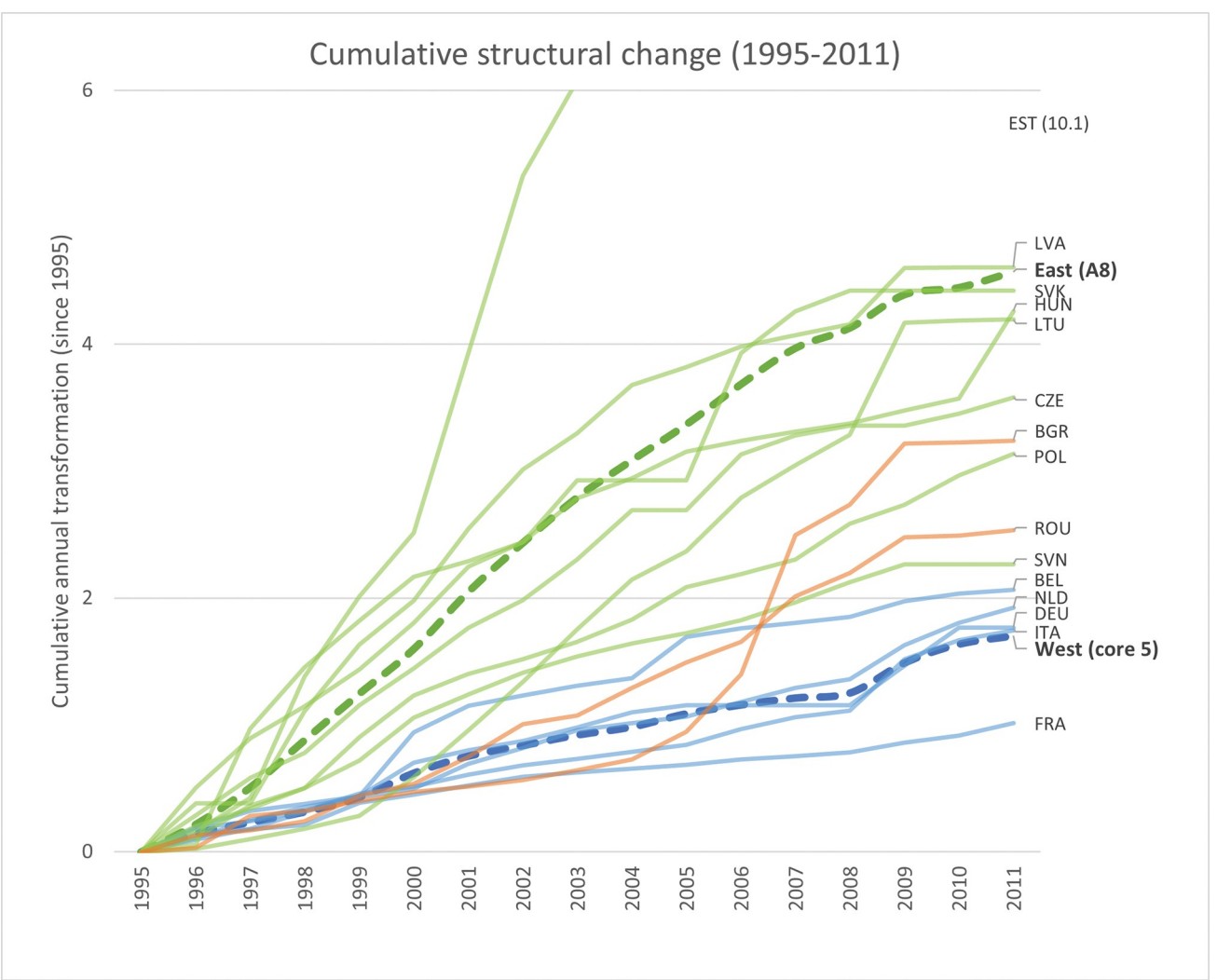

**Fig 13. Cumulative annual structural change of 10 Eastern and 5 Western economies over the 1995–2011 period.** Starting in 1995, each line corresponds to the cumulative annual transformation of each country in the analysis, separated by Western (blue) and Eastern (green for A8 and orange for Romania and Bulgaria) lines. The thicker, dashed lines represent regional average cumulative structural change for the five Western economies (blue) and the A8-group of Eastern economies (green).

dotted lines). Thus, production structures that might have been relatively similar in 1995, such as those of Poland and Hungary (0.30), and Latvia and Lithuania (0.34), ended up being quite dissimilar in 2011 (1.53 and 1.46, respectively). The corresponding trends in structural diversity among the 5 Western economies differ from their Eastern counterparts, where both the average pairwise dissimilarity (solid blue line in Fig 16) and the intra-Western structural variance (dashed blue lines) remain low in comparison.

To what extent did the Eastern structural transformation lead to a convergence or divergence with respect to the archetypal Western production structures? Examining the average pairwise dissimilarities between all Eastern and Western production structures, the average regional dissimilarity decreased only marginally from 1.49 to 1.34 –see solid orange line in Fig 17 below.

The structural trajectories of the Eastern economies thus seem most akin to an orbital trajectory: indeed on the move, yet remaining structurally equidistant from the seemingly

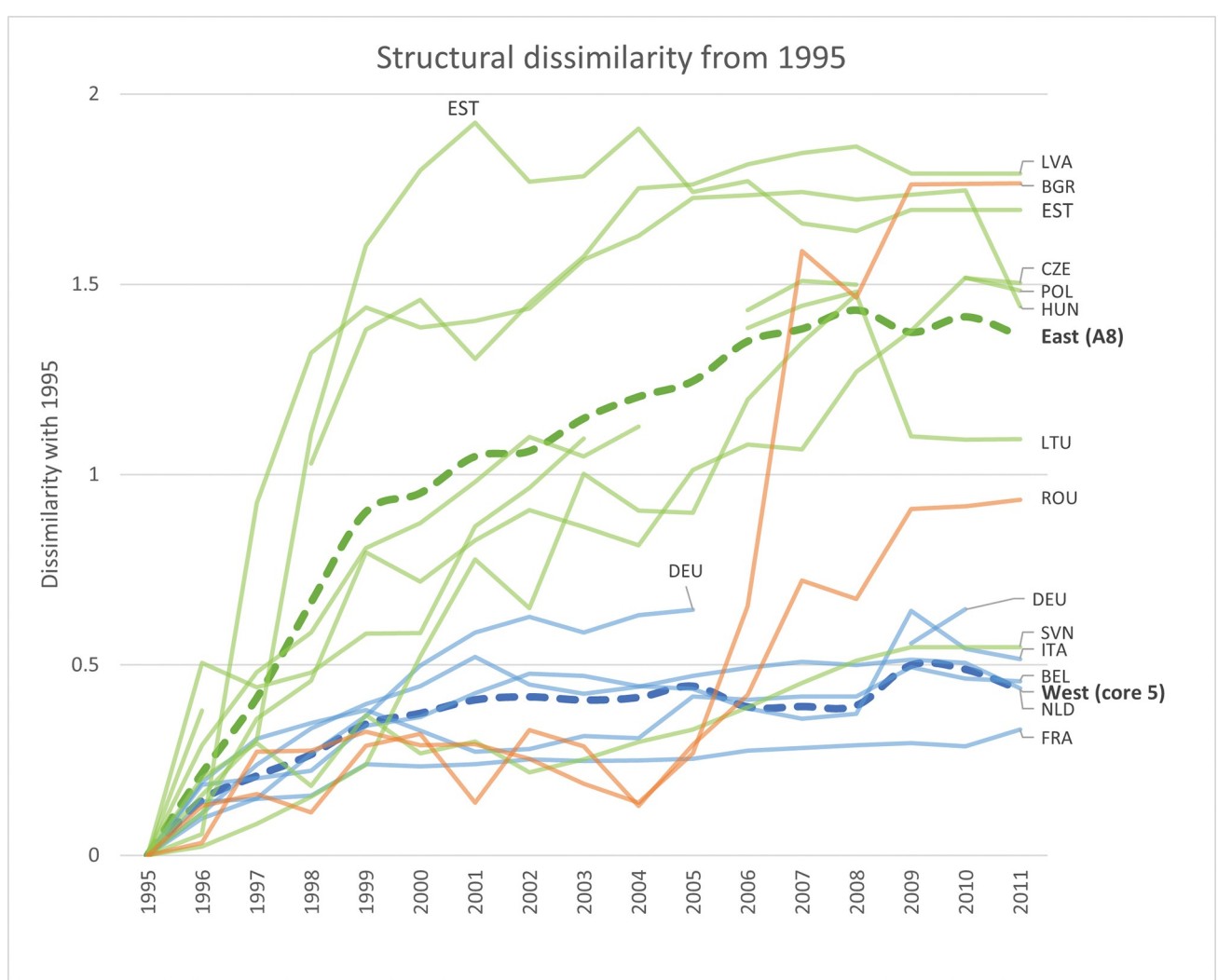

**Fig 14. Structural transformation of 8 Eastern and 5 Western economies during the period 1995–2011, as compared to their structural fingerprints in 1995.** Each line corresponds to the structural difference of each country with respect to its structural fingerprint in 1995, separated by Western (blue) and Eastern (green for A8 and orange for Romania and Bulgaria) lines. The thicker, dashed lines represent regional average structural differences for the five Western economies (blue) and the A8-group of Eastern economies (green), with respect to the index year 1995.

stationary state of Western production structures. (A supplementary analysis and visualization using multi-dimensional scaling on this set of EU countries–see Appendix D.2 –yields similar orbital findings). These findings indicate that East-West economic integration in terms of internal transformations of national economies exhibits a complexity and variability among the East- and Central-European countries, with potential implications for existing theories on East-West structural convergence [e.g. 33, 61, 62, 67, 68].

## 5. Summary and conclusion

This paper has proposed an eigenvector-based analytical framework for the comparative study of national production structures in the contemporary world of transnational production. Conceptualizing such structures as the complex networks of intra- and inter-sectorial flows between the domestic sectors of a national economy, including intermediate input originating from foreign sectors, the proposed approach utilizes both the national and international

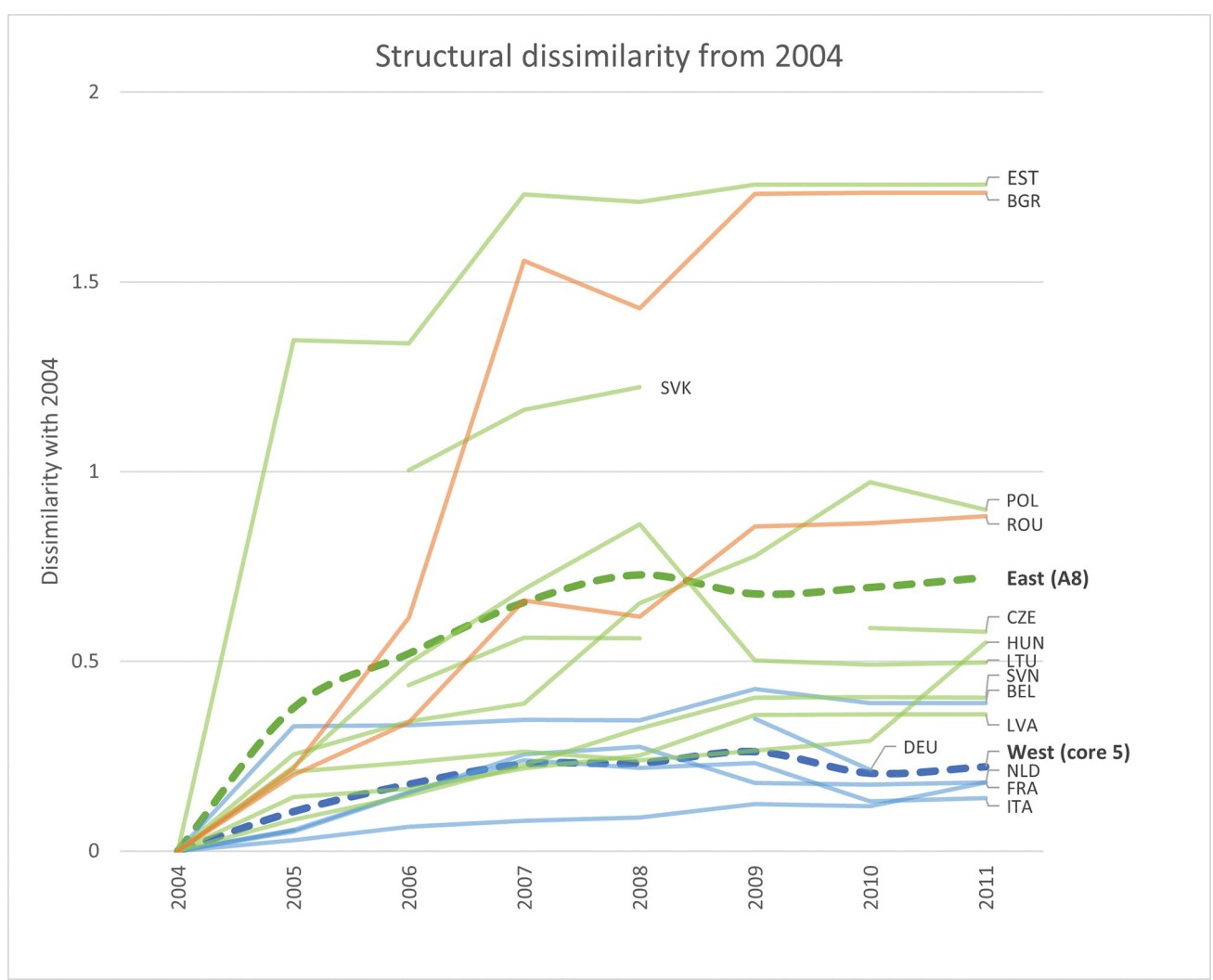

**Fig 15. Structural transformation of 8 Eastern and 5 Wester economies during the period 2004–2011, as compared to their structural fingerprints in 2004.** Each line corresponds to the structural difference of each country with respect to its structural fingerprint in 2004, separated by Western (blue) and Eastern (green for A8 and orange for Romania and Bulgaria) lines. The thicker, dashed lines represent regional average structural differences for the five Western economies (blue) and the A8-group of Eastern economies (green), with respect to the index year 2004. As Slovakia 2004 is removed from the analysis, its 2003 fingerprint is here used as the index.

components of multiregional input-output data to extract characteristic fingerprints of the production structures of countries. This allows for several novel types of spatiotemporal comparative analyses of the similarities, transformations, and trajectories of national economies in contemporary networks of global production.

Using the national input-output tables from WIOD (2013 release), covering 40 countries over 17 years, two case studies demonstrated the practical utility of the fingerprinting approach. The first case study explored similarity patterns in the full set of fingerprints using hierarchical clustering. From this, an 11-cluster partition was proposed as an analytically useful classification. Viewed longitudinally, several country-specific transitions between these 11 types seem to coincide with specific economic-historical events and policy shifts of respective country, supporting the potential usefulness of this classification and the fingerprint framework at large. Corresponding cluster analyses using gross sectorial exports found similarities

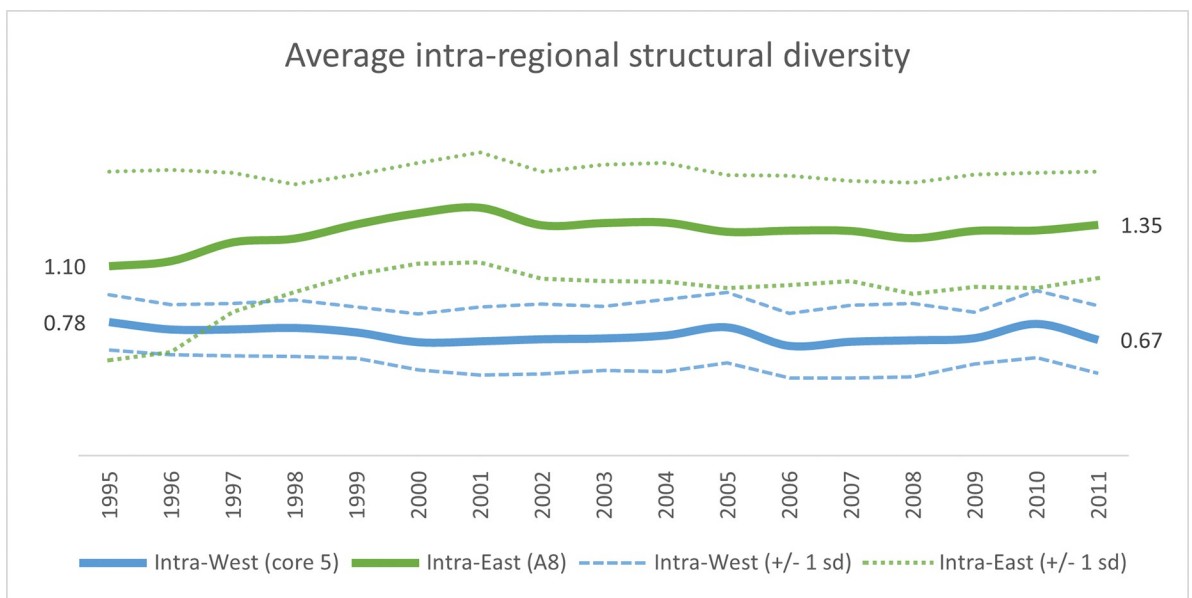

**Fig 16. Average within-group fingerprint dissimilarity among 8 Eastern and 5 Western economies over the period 1995–2011.** The solid green line captures the average structural dissimilarity among the A8-group of East-European economies, where the dashed green lines capture the within-group variance in terms of one standard deviation above and below the average. Corresponding structural dissimilarities for the five core EU countries are given by the solid and dashed blue lines.

that differed from the fingerprinting approach, underlining that the latter indeed captures properties that are not caught by examining gross sectorial export flows alone.

The second case study addressed East-West regional differences with regards to structural transformations and trajectories in the European Union in the period 1995–2011, contrasting

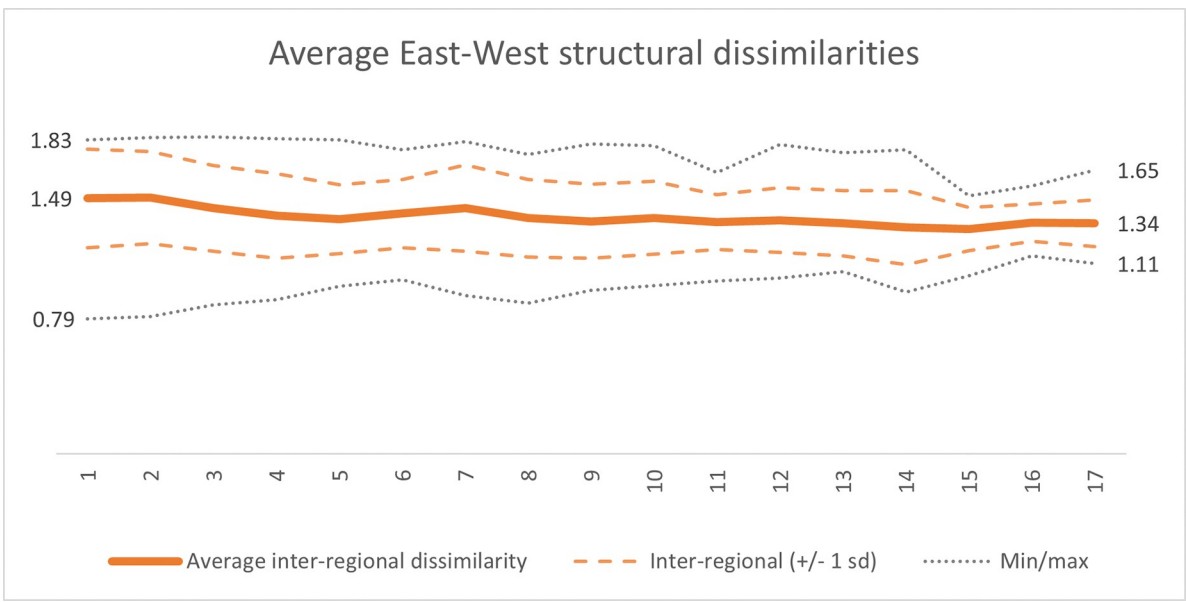

**Fig 17. Average fingerprint dissimilarity between the Eastern and Western countries over the period 1995–2011.** The solid orange line represents the average pairwise structural dissimilarity between all Western and Eastern economies in the analysis for each year, with the dashed orange line capturing variance in terms of one standard deviation above and below the mean. Minimum and maximum between-group dissimilarities are given by the black, dotted lines.

a Western set of five founding EU members with ten Central- and East-European countries that joined in 2004 and 2007. Finding significant structural transformation in the East during this period, the Western production structures changed much less. However, these large transformations of the Eastern economies had only a marginal effect on East-West production-structural convergence.

Despite the substantive findings from the two case studies, a certain degree of interpretational humility is warranted until the proposed approach has been validated and explored further. First, although the approach arguably produces feasible and seemingly interesting findings, the eigenvalue diagnostics for several of the WIOD13-derived fingerprints are low. In this study, a total of 27 country-year fingerprints were removed, but the specific threshold for such removals can indeed be debated. Although seemingly having little impact on obtained cluster-analytical results, interpretations of individual fingerprints with relatively poor eigenvalue diagnostics should be done with care. Second, whereas the approach has been tested here using the WIOD dataset (2013 release), it is imperative to also test the framework using other MRIO-type datasets and, specifically, compare the findings between these different datasets. (Whereas tentative analyses of the recently released OECD Inter-Country Input-Output Database (see http://oe.cd/icio) are promising, a more comprehensive analysis of this dataset, and comparison with the findings here, will have to wait for forthcoming studies).

Third, it is imperative that future evaluations of the specific fingerprinting operationalization should not only be conducted within the quantitative and computational domains, but more importantly within the qualitative and conceptual domains, to verify that the approach indeed captures features of relevance for the comparative study of national production structures in a transnational production regime. "Conceptions", as the saying goes, should indeed "precede and govern measurements" [72].

Finally, regardless of potential methodological improvements, it is imperative to view the proposed fingerprinting approach for what it is: an approach for the comparative analysis of national production structures. While it is arguably useful for understanding the structural transformations and trajectories of national economies in global production networks, the approach constitutes merely one of a plethora of different ways of conceptualizing what is meant by a 'production structure', both within and outside the realms of input-output data and quantitative analysis. Our contemporary global economy is indeed "an immensely complex, interdependent and dynamic system [and] our attempts to comprehend it analytically are always partial, provisional and incomplete" [8, see also 73]–it is hoped that the fingerprinting framework can provide yet another incomplete, partial, and complementary way of understanding this complexity.

## Supporting information

**S1 Appendix. Appendixes: Transformations, trajectories, and similarities of national production structures.** Tables, metadata and two additional fingerprinting analyses pertaining to the article.
(PDF)

## Author Contributions

**Conceptualization:** Carl Nordlund.

**Data curation:** Carl Nordlund.

**Formal analysis:** Carl Nordlund.

**Funding acquisition:** Carl Nordlund.

**Investigation:** Carl Nordlund.

**Methodology:** Carl Nordlund.

**Project administration:** Carl Nordlund.

**Resources:** Carl Nordlund.

**Software:** Carl Nordlund.

**Validation:** Carl Nordlund.

**Visualization:** Carl Nordlund.

**Writing – original draft:** Carl Nordlund.

**Writing – review & editing:** Carl Nordlund.

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
