## [Decision Letter · Decision Letter 0]

4 Sep 2023

PONE-D-23-16568Transformations, trajectories, and similarities of national production structures: A comparative fingerprinting approachPLOS ONE

Dear Dr. Nordlund,

Thank you for submitting your manuscript to PLOS ONE. After careful consideration, we feel that it has merit but does not fully meet PLOS ONE’s publication criteria as it currently stands. Therefore, we invite you to submit a revised version of the manuscript that addresses the points raised during the review process.

The reviewers agree that the paper have merits and could be an important addition to the field. However, they both raise concerns regarding the relation between the theoretical framework and the analysis. I suggest you carefully implement the suggestions of both reviewers and explain more clearly your methodological considirations.

We look forward to receiving your revised manuscript.

Kind regards,

Eyal Bar-Haim

Academic Editor

PLOS ONE

Journal Requirements:

“This research was partly supported by NordForsk through the funding to The Network Dynamics of Ethnic Integration, project number 105147, the Swedish Research Council (DNR 445-2013-7681), and Budapest Közép-Európai Egyetem Alapitvány (CEU BPF). The funders had no role in study design, data collection and analysis, decision to publish, or preparation of the manuscript.”

3. Please remove your figures from within your manuscript file, leaving only the individual TIFF/EPS image files, uploaded separately. These will be automatically included in the reviewers’ PDF.

Reviewers' comments:

Reviewer's Responses to Questions

**Comments to the Author**

1. Is the manuscript technically sound, and do the data support the conclusions?

Reviewer #1: Yes

Reviewer #2: Yes

2. Has the statistical analysis been performed appropriately and rigorously? 

Reviewer #1: Yes

Reviewer #2: Yes

3. Have the authors made all data underlying the findings in their manuscript fully available?

Reviewer #1: Yes

Reviewer #2: Yes

4. Is the manuscript presented in an intelligible fashion and written in standard English?

Reviewer #1: Yes

Reviewer #2: Yes

5. Review Comments to the Author

Reviewer #1: This paper reports on an original and innovative work and is a part of a larger research project.

The paper suggests a network-analytical framework for the comparative study of national

21 production structures in global production networks, using input-output tables (from national account statistics), attempting at extractIng a structural profile, or a “structural signature”, deem capturing the up- and downstream prominence of economic sectors for a particular country and 25 years. These ‘fingerprints’ of national production structures can subsequently be compared on a pairwise comparison of the structural similarities, transformations, and trajectories of national economies in the transnational production regime.

The work is innovative, informative, useful, and timely. I enjoyed it, learned a lot, and appreciated the research agenda and methodological originality.

The specified network indices and derivative measures are performed to high technical standards and are described in sufficient detail.

.

The conclusions are presented in an appropriate and eloquent fashion and are supported by the data and the Appendices.

The article is presented in an intelligible fashion and is written in standard English.

The author provides clearly stated explanations, examples, illustrations and tables, also providing specific details regarding data availability. All data used in this draft is publicly available for download at the project website

My only minor suggestions are:

1. What is the difference between Dissimilarity index and the known measure of Structural Equivalence (measured by Euclidean Distances See Burt 1988, Burt and Carlton, 1989, who also analyzed Input-Output tables),

Burt, R. S. (1988). ‘‘The Stability of the American Markets,’’ American Journal of Sociology 94, 356–95.

Burt, R. S. and Carlton, D. (1989). ‘‘Another look at the network boundaries of American markets,’’American Journal of Sociology 95, 723–753.

2. Why not produce a matrix of Similarity Index between all or most of the data points? Is it possible then to extend the method to include more than merely pairwise comparison? (The matrix can be compacted into simple map by data reduction techniques),

3. What about the relations between structural dominance and performance?

Other works have linked network analytic tools of I/O, derived power position, and dependent variables (mostly profitability). See, for example, Talmud, I., “Relations and Profits: Imperfect Competition and Its Outcome”, Social Science Research, 23, 1994: 109-135. Talmud, I., “Industry Market Power, Industry Political Power, and State Support: The Case of Israeli Industry”, Research in Politics and Society, 4: 35-62, 1992.

4. What is the primacy of the “fingerprint” approach to external network analyses of the world system?

See:

Van Rossem, Ronan. "The world system paradigm as general theory of development: A cross-national test." American sociological review (1996): 508-527.‏

Snyder and Kick, 1979

(and also Clark, 2010; Clark and Beckfield, 2009; Clark and Mahutga, 2013; Kick and Davis, 2001; Kick, McKinney, McDonald, and Jorgenson, 2011; Lloyd et al., 2009; Mahutga, 2006; Mahutga and Smith, 2011; Nemeth and Smith, 1985; Smith and White, 1992)

Z. Maoz. R. Kuperman, L. Terris., and I. Talmud (2006). “Structural Equivalence and International Conflict, 1816-2000: A Social Networks Analysis of Dyadic Affinities and Conflict.”. Journal of Conflict Resolution 50: 664-689).

5. Table 3 should be moved to the Appendices.

Reviewer #2: Review: Transformations, trajectories, and similarities of national production structures

The author suggests a framework for the comparative assessment of national production structures via the spectral analysis of Countries’ input and output proposed by Dietzenbacher. In particular, the author employs the right-hand dominant eigenvector of the intermediate-use matrix Z and the left-hand dominant eigenvector of the matrix T = Z + M as proxies for the timely downstream and upstream prominence of a country in a given sector. According to the author, the two indices can be used to compare countries in terms of production structure and to analyse their development through time. The authors provide also two case studies:

1) A cluster analysis on the distance matrices of countries' “finger prints”,

2) a focus on the relative development of the Eastern European countries w.r.t. the Western European.

The work represents an advanced application of the methodology proposed by Dietzenbacher. The analysis is interesting, although the degree of novelty with respect to the current literature should be stated more clearly. Overall, the work is structured and well written. Nonetheless, some points require more attention:

MAJOR

- Please improve the discussion concerning the degree of novelty of the proposed framework in light of the current literature. You may want to consider also the following articles

Giammetti, R., Russo, A., & Gallegati, M. (2020). Key sectors in input–output production networks: An application to Brexit. The World Economy, 43(4), 840-870.

Wang, X., Wang, Z., Cui, C., & Wei, L. (2020). Forward and backward critical sectors for CO 2 emissions in China based on eigenvector approaches. Environmental Science and Pollution Research, 27, 16110-16120.

Liu, E., & Tsyvinski, A. (2020). Dynamical structure and spectral properties of input-output networks (No. w28178). National Bureau of Economic Research.

Cerina, F., Zhu, Z., Chessa, A., & Riccaboni, M. (2015). World input-output network. PloS one, 10(7), e0134025

- Up-stream and down-stream “finger prints” seem to be fairly correlated to one-another. One may wonder whether the results may be severely altered if just one of the two indices is considered in computing the Euclidean distance (Eq. 3). Similarly, the author may want to test other ways to compare two vectors. For instance, the author might consider using cosine similarity emphasizing similarities and differences w.r.t. the Euclidean distance already used.

- Regarding the second case study, the author may also be willing to consider a deeper discussion of his findings in light of the economic theory and the recent history of the Eastern European countries, which appears to be missing. Moreover, the analysis performed by the author may benefit from a ranking of Eastern European countries according to their degree of convergence to the Western countries' fingerprints.

MINOR

Captions should be improved as it needs to be self-contained. For what concerns Figure 1, please provide a short description of the different nodes.

Different font in “Dietzenbacher’s” , l. 240

Please clarify in the text the use of the symbol ⊕ as it might be ambiguous. 292.

Please clarify according to which criterion “…the mean and median diagnostics are satisfactory”, l.356.

6. PLOS authors have the option to publish the peer review history of their article (what does this mean?). If published, this will include your full peer review and any attached files.

Reviewer #1: No

Reviewer #2: No

---

## [Author Response · Author response to Decision Letter 0]

8 Nov 2023

See document 'Response to reviewers.docx' for my response to reviewers

---

## [Decision Letter · Decision Letter 1]

27 Nov 2023

Transformations, trajectories, and similarities of national production structures: A comparative fingerprinting approach

PONE-D-23-16568R1

Dear Dr. Nordlund,

We’re pleased to inform you that your manuscript has been judged scientifically suitable for publication and will be formally accepted for publication once it meets all outstanding technical requirements.

Kind regards,

Eyal Bar-Haim

Academic Editor

PLOS ONE

Additional Editor Comments (optional):

Reviewers' comments:

Reviewer's Responses to Questions

**Comments to the Author**

1. If the authors have adequately addressed your comments raised in a previous round of review and you feel that this manuscript is now acceptable for publication, you may indicate that here to bypass the “Comments to the Author” section, enter your conflict of interest statement in the “Confidential to Editor” section, and submit your "Accept" recommendation.

Reviewer #2: All comments have been addressed

2. Is the manuscript technically sound, and do the data support the conclusions?

Reviewer #2: Yes

3. Has the statistical analysis been performed appropriately and rigorously? 

Reviewer #2: Yes

4. Have the authors made all data underlying the findings in their manuscript fully available?

Reviewer #2: No

5. Is the manuscript presented in an intelligible fashion and written in standard English?

Reviewer #2: Yes

6. Review Comments to the Author

Reviewer #2: The authors addressed all points of the report. Please double check with them the availability of data and code

7. PLOS authors have the option to publish the peer review history of their article (what does this mean?). If published, this will include your full peer review and any attached files.

Reviewer #2: No

---

## [Editor Report · Acceptance letter]

6 Dec 2023

PONE-D-23-16568R1 

Transformations, trajectories, and similarities of national production structures: a comparative fingerprinting approach 

Dear Dr. Nordlund:

I'm pleased to inform you that your manuscript has been deemed suitable for publication in PLOS ONE. Congratulations! Your manuscript is now with our production department. 

Kind regards, 

on behalf of

Dr. Eyal Bar-Haim 

Academic Editor

PLOS ONE